# On the convection of ionospheric density features

John D. de Boer[1], Jean-Marc A. Noël[1], and Jean-Pierre St.-Maurice[2]

[1]Royal Military College, P.O. Box 17000, Kingston, ON, Canada
[2]Institute of Space and Atmospheric Studies, University of Saskatchewan, Saskatoon, SK, Canada

**Correspondence:** John D. de Boer (john.deBoer@rmc.ca)

**Abstract.** We investigate whether the boundaries of an ionospheric region of different density than its surroundings will drift relative to the background $\boldsymbol{E} \times \boldsymbol{B}$ drift and, if so, how the drift depends on the degree of density enhancement and the altitude. We find analytic solutions for discrete circular features in a 2-D magnetised plasma. The relative drift is proportional to the density difference, which suggests that where density gradients occur they should tend to steepen on one side of a patch while they are weakened on the other. This may have relevance to the morphology of polar ionospheric patches and auroral arcs, since the result is scale-invariant. There is also an altitude dependence which enters through the ion-neutral collision frequency. We discuss how the 2-D analytic result can be applied to the real ionosphere.

## 1 Introduction

We are investigating the fundamental transport properties of cold, magnetised plasmas. There appears to be a characteristic property of $\boldsymbol{E} \times \boldsymbol{B}$ drift which has not been previously elucidated and which we examine here. The expression for this drift field $\mathbf{v}_a$ (subscript *a* for *ambipolar*) is

$$\mathbf{v}_a = \frac{\boldsymbol{E} \times \boldsymbol{B}}{B^2} \tag{1}$$

which does not involve either mass or number density, or indeed *any* of the intrinsic properties of the plasma. The phenomenon of ambipolar drift can be understood either from the trochoidal trajectory of individual particles, or from the Lorenz transformation, which shows that for a magnetised plasma there is a "preferred" frame of reference (this $\mathbf{v}_a$) in which the perpendicular electric field $\boldsymbol{E}_\perp$ vanishes.

The $\boldsymbol{E} \times \boldsymbol{B}$ drift is independent of the plasma properties – *for a given $\boldsymbol{E}$*. However we may ask how the electric field will be arranged in and around a plasma density feature. We show that $\boldsymbol{E}$ can and will likely be structured in such a way that $\boldsymbol{E} \times \boldsymbol{B}$ drift *does* depend inversely on the plasma mass density in most situations. The conditions under which our initial assumptions prevail in the ionosphere and magnetosphere will also be discussed.

The analysis we present entails certain simplifying assumptions, the chief of which is uniformity along magnetic field lines, also called a 2-D plasma. More precisely, it means that the location of plasma *along* a field line is considered unimportant,

so that we need to consider only height-integrated (or field-line averaged) plasma properties. This assumption is clearly a significant one in the ionosphere, especially in the E-region. However the high parallel conductivity at all altitudes causes magnetic field lines to have a consistent electric potential, which forces the $\boldsymbol{E} \times \boldsymbol{B}$ drift to be mapped along the entire flux tube (propagated at the Alfvén speed). We can also show that on *closed* field lines our results are still applicable to an ionospheric situation. We believe that numerical modelling with realistic field-line gradients will also substantiate this analysis.

We discuss electric field rearrangements, denoted $\delta\boldsymbol{E}_{\perp}$, which require finite time to propagate through the 2-D domain perpendicular to $\boldsymbol{B}$. However the low-frequency (DC) limit for *perpendicular* EM propagation in the ionosphere is also the Alfvén speed, $v_{A}$, as shown e.g. by Baumjohann and Treumann (1996, Eq. 9.142). Typical ambipolar drifts are *much* slower, so one can assume that $\delta\boldsymbol{E}_{\perp}$ propagates effectively instantaneously.

We call the average electric field far from the density features we consider the "background" electric field, $\boldsymbol{E}_0$. As we shall see below, the background field can only be uniform if the background density is as well.

If we were to address *collisionless* plasma, i.e. with no significant ion-neutral collisions, there is a train of argument we can take to show how the electric field becomes structured merely by propagating through density features. The result one obtains is the same as we obtain below, if we then look at the limit of collision frequency approaching zero; the time scale is finite and dependent on propagation, not collisions. However this case entails as much work as this already lengthy article, and will hopefully be the subject of a follow-on paper. The results below are more than adequate for any *ionospheric* conditions.

In this treatment, we assume that there are initally no field-aligned currents (FACs) within our domain of interest. Instead the electric field, and associated plasma drift, occur because of driving forces *outside* our domain. In a region of *open* field lines, or where the plasma is being forced *locally* by the neutral dynamo, the drift speed could remain independent of local density, e.g. the polar-cap potential field is "mapped" there more or less directly from the interplanetary electric field. But it still creates an electric field on *closed* field lines in the polar regions. The scenarios we study are an idealisation of conditions in the polar region, but as long as there is an electric field in the frame of the neutrals our results apply in some measure.

We shall examine a circular plasma density feature, either an enhancement or a depletion. For ease of reference we use the word "patch" in this paper, but with that term we don't mean *only* features that are two times or greater in density than the background, which is its conventional definition (e.g. Carlson, 2012).

Let the density of the patch be $n$ times the background density. Thus a depletion is a density feature with $0 < n < 1$. For ease of later notation, we find it useful to define another dimensionless quantity:

$$\eta = \frac{n-1}{n+1} \tag{2}$$

Thus $\eta$ serves as an alternate parameterisation of relative density such that $-1 < \eta < 1$, the bounds corresponding to an extreme depletion ($\eta \to -1$ as $n \to 0$) or an extremely dense patch ($\eta \to 1$ as $n \to \infty$).

## 1.1 Boundary condition on moving, sharp interface

In studying a scenario with varying density and drift speed, we must maintain conservation of particle number by species. This leads to a constraint on the speed at which the *boundary* drifts.

Let $\hat{\boldsymbol{n}}$ be an outward normal, and let $\rho_s$ be the density of species $s$. If we call the drift of the sharp, step boundary $\mathbf{v}_\mathrm{b}$ then conservation of particle number requires that at the interface, for each species,

$$\rho_s \hat{\boldsymbol{n}} \cdot (\mathbf{v}_s - \mathbf{v}_\mathrm{b})\Big|_\mathrm{ext} = \rho_s \hat{\boldsymbol{n}} \cdot (\mathbf{v}_s - \mathbf{v}_\mathrm{b})\Big|_\mathrm{int} \tag{3}$$

which we can rearrange to get

$$\hat{\boldsymbol{n}} \cdot \mathbf{v}_\mathrm{b} = \frac{\hat{\boldsymbol{n}} \cdot (n\mathbf{v}_\mathrm{s,int} - \mathbf{v}_\mathrm{s,ext})}{(n-1)} \tag{4}$$

Also, the component of $\mathbf{v}_\mathrm{b}$ tangential to the boundary is arbitrary. So if we obtain a result like Eq. (4) where the bracketed expression on the RHS has no angular dependence, then we can drop $\hat{\boldsymbol{n}}\cdot$ from both sides of the last equation and choose the vector $\mathbf{v}_\mathrm{b}$ to be $(n\mathbf{v}_\mathrm{s,int} - \mathbf{v}_\mathrm{s,ext})/(n-1)$.

Now, Eq. (4) in this model does not include terms for ionisation and recombination, which are significant in the E region.

At the boundary *itself*, these terms might not be important compared to the flux of particles through the sharp boundary in an arbitrarily short time, but $n$ is different on each side, so the ionisation/recombination balance must be different; and since the production mechanism cannot be assumed to drift with the patch, the results we obtain are valid only on the timescale in which the original density difference remains significant under ambient production.

There are further objections that could be raised to studying sharp boundaries, but we address those challenges in the

15 Discussion. It might also be questioned why particle number but not *momentum density* appears to be conserved at the boundary, and we explain this in App. C.

We also wish to draw the reader's attention to the fact that we present only a *steady-state* picture: the generation of small-scale potential features in the ionosphere that are different from the large-scale convection pattern must trigger Alfvén waves that transport small-scale stresses back up from the ionosphere to the magnetosphere. The propagation timescales and the

20 momentum reserves in the magnetosphere are beyond the scope of this study. We only posit that both are finite, so that there must be some conditions under which the present model has at least limited application. This caveat is essentially the same as the earlier warning that our results are only strictly appropriate for a 2-D plasma, and only applicable to the extent that a plasma has a 2-D character.

The first-order effect of currents and charge accumulations in the ionosphere is a $\delta\boldsymbol{B}$ which launches an Alfvén wave that

establishes an FAC. But the net effect is a return to current closure, i.e. $\nabla \cdot \boldsymbol{J} = 0$, with a modified $\boldsymbol{E}$. Our analysis strides over those processes and looks at the net result. Vasyliūnas (2005) offers an explanation in his Sec. 3.2, especially its last paragraph. Both the large-scale FACs that we assume are present *outside* our domain, and the small-scale FACs that our model would implicitly trigger but which we *ignore*, couple the magnetosphere to the ionosphere. The net, time-dependent convection would be quite complicated. Here we are endeavouring to show *only* what form the ionosphere's side of the forcing terms might

take.

## 1.2 Argument for 2-D assumption

Let us consider a 2-D plasma with collisions with a neutral gas, like the E-region but isolated from any parallel connections with a higher, collisionless region. It is uniform in the parallel direction, but we shall relate it to a realistic scenario in the

Discussion. There is a background electric field in the +y direction. The Pedersen and Hall currents will be assumed to be driven by – and to close with – parallel currents far off in the ±y directions.

The FACs have their origin in plasma drifts in the magnetosphere or, in the case of open field lines, in the solar wind. FACs will create or adjust the ionospheric electric field whenever the ionospheric drift is not coherent with the magnetospheric drift: that is, the electric potential is mapped along field lines.

If E-region conductivity features begin to structure $E_\perp$, then initially the F-region will create FACs that reduce the trend. However in App. B we show that the F-region's drift kinetic energy would be absorbed by the E-region in a time on the order of a second or less, leading the F-region to adopt the E-region's $E_\perp$ structure also.

Then, once the whole height of the ionosphere has developed a structured $E_\perp$, the magnetosphere's much greater drift kinetic energy reserve will continue to feed FACs structured in such a way as to restore the ionosphere's drift to the pattern of the magnetosphere's drift.

However, the magnetosphere is far enough from the ionosphere that it cannot provide FACs immediately, once the drift kinetic energy of the F-region has been used up by E-region conductivity. The time for an Alfvén wave to travel to the magnetic equator and back is of order 3 to 5 minutes, and there is only a finite energy available even there. Therefore it is reasonable to consider that most of the auroral ionosphere is typically in a condition where the electric field re-arrangements necessary for the E-region to be in steady state ($\nabla \cdot \boldsymbol{J} = 0$) have been more or less mapped up throughout the F-region as well. (And the E-region would ultimately impose its electric-field structuring upwards onto the remainder of any *closed* field line, given sufficient time.) Moreover, the E-region's conductivity structure will generate $E_\perp$ structure within the time scales of the ion-neutral collision frequency $\nu_\mathrm{in}$ and of the perpendicular EM propagation, which is also at the Alfvén speed.

Therefore we feel confident in proceeding with an analysis that excludes FACs within the domain of interest, and in which the electric field in the E-region is determined only by the "background" $E_0$ and by E-region conductivity.

## 1.3   E-region steady state without local FAC

In a uniform 2-D plasma the current density is

$$\boldsymbol{J} = [\sigma]\boldsymbol{E} = \begin{bmatrix} \sigma_\mathrm{P} & -\sigma_\mathrm{H} \\ \sigma_\mathrm{H} & \sigma_\mathrm{P} \end{bmatrix} \boldsymbol{E} \tag{5}$$

where $\sigma_\mathrm{P}$ and $\sigma_\mathrm{H}$ are the Pedersen and Hall conductivities. (We use the letter $\sigma$ elsewhere for a surface-charge density, so to avoid confusion we write the conductivity matrix as $[\sigma]$.) The components are equal to

$$\sigma_\mathrm{P} = \sum_s \frac{q_s n_s \kappa_s}{B(1 + \kappa_s^2)} \tag{6}$$

$$\sigma_\mathrm{H} = \sum_s \frac{q_s n_s}{B(1 + \kappa_s^2)} \tag{7}$$

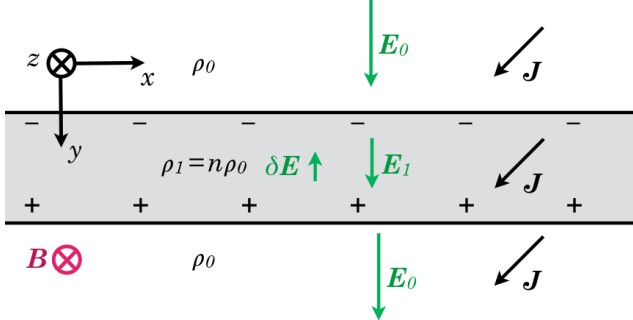

**Figure 1.** A slab of denser plasma extending perpendicular to the background electric field $\boldsymbol{E}_0$ and extending parallel to $\boldsymbol{B}$ (e.g. between two L-shells). In order for the Pedersen current across the boundaries to be conserved, $\boldsymbol{E}_1$ within the slab will be lower than $\boldsymbol{E}_0$ outside.

where the sum is over all charged species; $q_s$, $n_s$ and $\kappa_s$ are the charge, number density and magnetisation ratio of the species $s$. The latter is defined as $\kappa_s = \omega_s / \nu_s$, where $\omega_s$ is the cyclotron frequency of the species and $\nu_s$ is its momentum transfer collision frequency with the neutrals. (We consider both $\omega_\mathrm{e}$ and $\kappa_\mathrm{e}$ to be negative.)

We denote quantities within the enhancement with a prime. If the patch is a factor $n$ times the number density of the background, assuming similar composition, then $[\sigma]' = n[\sigma]$.

## 2    A "slab" feature in the E-region

Let us first consider a slab geometry. Assume there is a curtain of plasma as in Fig. 1 with different density than its surroundings. (By *slab* we mean something like different L shells, not *horizontal* strata unless we were at the equator.) We assume Cartesian geometry with $\boldsymbol{B} = B\hat{\boldsymbol{z}}$.

We say E-region because we are taking account of the Hall current density. The results for both the slab and the circular geometry are equally valid for F-region patches, with the considerations explained in the Discussion.

If $\boldsymbol{E}$ were uniform, the higher conductivity inside the slab would build up charge on one boundary and deplete it on the other, creating a $\delta\boldsymbol{E}$ oriented towards $-\hat{\boldsymbol{y}}$ and reducing $E$ within the slab. In steady state, $\delta\boldsymbol{E}$ will be the value that restores $\nabla \cdot \boldsymbol{J}_\perp = 0$.

We can see by inspection that $E' = E_0 + \delta E$ must be inversely proportional to the density ratio $n$, since the Pedersen current density has to be equal on both sides of the boundary. Provided $\nu_\mathrm{in}$ is constant, the Hall current density remains uniform as well, since the product $E'\sigma'_\mathrm{H}$ is concomitantly fixed. The Pedersen component of ion drift is in the $+y$ direction, faster outside the layer than within it. This means that ions are piling up on the incoming side of the layer and being peeled away on the other. The net result is that the boundary between the more and less dense regions does *not* move, and that the *ions* within the slab are transient while the electrons remain in it.

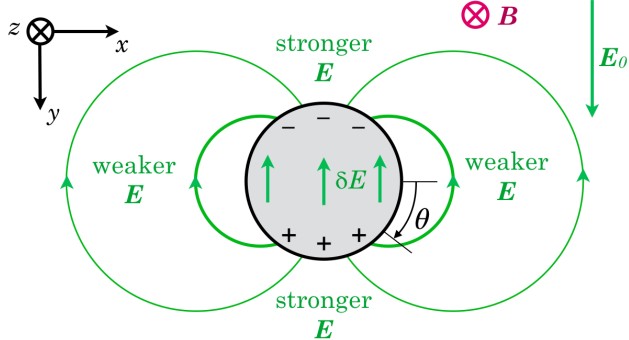

**Figure 2.** A circular patch of denser plasma will acquire a polarisation that reduces the electric field strength inside compared to the background field strength. This sets up a cylindrical (2-D) dipole. The $+$ and $-$ show net surface charge. The sense of the angle $\theta$ used in the analysis is shown (the conventional sense, but appearing clockwise because $\hat{z}$ is into the page). This figure ignores Hall current, but it is included in our analysis, and it is shown in Fig. 4. The sense of plasma drift around such a feature is sketched in Fig. 3.

## 3 A circular, E-region patch

Suppose next that there is a circular patch of higher density, as suggested in Fig. 2. The electric field strength $E$ must be lower inside the patch, or else charge would continually build up at the boundary. In fact in steady state there must be a net charge on the interface in order for $E$ to be lower inside. By inspection or simple argument we can see that it will have a cylindrical
dipole arrangement.

This problem has been solved already by Hysell and Drexler (2006) using complex analysis, and they have even obtained the solution for a more general elliptical problem using a conformal mapping. Up to Eq. (27) we provide an alternate derivation of the same result. Our method for the circular patch is somewhat less abstract, and perhaps simpler, because it does not use the double shell required for their elliptical result.
The cylindrical (or 2-D) dipole has a distinct character from the spherical dipole that is more familiar in space physics contexts. Using cylindrical polar coordinates $\rho$ and $\theta$, the components of a 2-D dipole aligned with the x axis are (Mallinson, 1981, Eq. 8)

$$a_\rho = \frac{2\mu}{\rho^2}\cos\theta; \; a_\theta = \frac{2\mu}{\rho^2}\sin\theta \tag{8}$$

where $\mu$ is a constant and $\boldsymbol{a} = (a_\rho, a_\theta)$ represents the dipole field. Both the field lines and equipotential surfaces of a 2-D dipole
are circular. So the disturbed velocity field is also a dipole rotated by $90°$ as seen in Fig. 3. In App. A we provide a Cartesian expression for the dipole and details of the algebraic steps.

In Figs. 2 and 3 the dipole is aligned with the y axis, and the net charge along the circular boundary has the form $\sigma = \sigma_0\sin\theta$; the sense of $\theta$ is shown in Fig. 2. Let $\boldsymbol{E}_{\text{dip}}$ be a vector oriented parallel to the charge dipole and representing its maximum strength; then $\sigma_0 = 2\varepsilon_0 E_{\text{dip}}$. The electric field in and around the patch has the form

$$\boldsymbol{E}_{\text{int}} = \boldsymbol{E}_0 - \boldsymbol{E}_{\text{dip}} \tag{9}$$

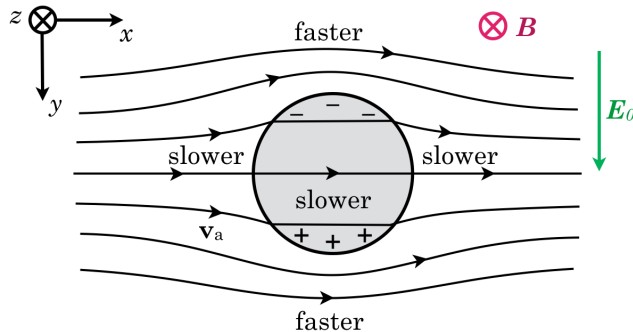

**Figure 3.** The polarisation of the patch in Fig. 2 establishes a dipole disturbance in the $\boldsymbol{E} \times \boldsymbol{B}$ drift. Note that plasma flows through the patch: while the density enhancement should retain its circular shape, the constituent ions are transient.

$$\boldsymbol{E}_{\text{ext}} = \boldsymbol{E}_0 + \frac{R^2}{\rho^2} D(\theta) \boldsymbol{E}_{\text{dip}} \tag{10}$$

where $R$ is the radius of the patch and $D$ is a matrix defined in App. A.

When we take account of the Hall conductivity, we find that the dipole is no longer oriented exactly in the $+y$ direction. Nor can $\boldsymbol{E}$ and $\boldsymbol{J}$ inside the patch be simply parallel, but scaled-down versions of their values outside the patch as they are for the slab, because the Hall current would then accumulate on the interface as a dipole oriented towards $-x$. The angle which $\boldsymbol{J}$ makes with $\boldsymbol{E}$ means that the positive pole of the dipole will be shifted from 6 o'clock in that figure toward 7 o'clock. So the field inside the patch, $\boldsymbol{E}_{\text{int}}$, will be oriented somewhat towards 5 o'clock. See Fig. 4. If we assume a dipole orientation angle as a variable we can solve for both it and the dipole strength by requiring the current density expressions on the inner and outer sides of the circular boundary to be identical. (It is something like getting a doughnut and its hole to travel at the same speed.) Using the two mentioned parameters, this condition can be satisfied.

With $\boldsymbol{E}_0 = E_0 \hat{\boldsymbol{y}}$ we assume the perimeter of the patch develops a surface charge density in the form of a rotated dipole:

$$\sigma_{\text{net}} = \sigma_x \cos\theta + \sigma_y \sin\theta \tag{11}$$

We solve for $\sigma_x$ and $\sigma_y$ under the condition that the current $\boldsymbol{J}$ into the boundary from the inside must balance the current on the outside. The outward normal is $\hat{\boldsymbol{n}} = [\cos\theta, \sin\theta]^T$.

### 3.1 Steady state currents around a circular, E-region patch

Using the result of App. A and the definition in Eq. (9), $\delta\boldsymbol{E}_{\text{int}} = -\boldsymbol{E}_{\text{dip}}$. For sake of brevity we let $\boldsymbol{k} = \boldsymbol{E}_{\text{dip}}$ for the coming passage up to Eq. (20).

$$\boldsymbol{J}_{\text{int}} = [\sigma]' \begin{bmatrix} -k_x \\ E_0 - k_y \end{bmatrix} \tag{12}$$

The current into the inner side of the boundary is

$$
\begin{aligned}
J_{\text{int}} &= \begin{bmatrix} -k_x\sigma_{\text{P}} - E_0\sigma_{\text{H}} + k_y\sigma_{\text{H}} \\ -k_x\sigma_{\text{H}} + E_0\sigma_{\text{P}} - k_y\sigma_{\text{P}} \end{bmatrix}' \cdot \hat{\boldsymbol{n}} \\
&= \cos\theta(-k_x\sigma_{\text{P}} - E_0\sigma_{\text{H}} + k_y\sigma_{\text{H}})' \\
&\quad + \sin\theta(-k_x\sigma_{\text{H}} + E_0\sigma_{\text{P}} - k_y\sigma_{\text{P}})'
\end{aligned}
\tag{13}
$$

5 Using another result of App. A, at $\rho = R$,

$$
\delta\boldsymbol{E}_{\text{ext}} = (k_x\cos\theta + k_y\sin\theta)\hat{\boldsymbol{\rho}} + (k_x\sin\theta - k_y\cos\theta)\hat{\boldsymbol{\theta}}
\tag{14}
$$

$$
\boldsymbol{J}_{\text{ext}} = [\sigma]\begin{bmatrix} k_x\cos^2\theta + 2k_y\sin\theta\cos\theta - k_x\sin^2\theta \\ E_0 + 2k_x\sin\theta\cos\theta + k_y\sin^2\theta - k_y\cos^2\theta \end{bmatrix}
\tag{15}
$$

which after some straightforward steps yields a current *out of* the boundary of

10 $J_{\text{ext}} = E_0(\sigma_{\text{P}}\sin\theta - \sigma_{\text{H}}\cos\theta) + k_x\sigma_{\text{P}}\cos\theta$

$$
- k_x\sigma_{\text{H}}\sin\theta + k_y\sigma_{\text{P}}\sin\theta + k_y\sigma_{\text{H}}\cos\theta
\tag{16}
$$

Setting the cosine terms in $J_{\text{int}}$ and $J_{\text{ext}}$ equal,

$$
(\sigma_{\text{P}} + \sigma_{\text{P}}')k_x + (\sigma_{\text{H}} - \sigma_{\text{H}}')k_y = E_0(\sigma_{\text{H}} - \sigma_{\text{H}}')
\tag{17}
$$

and setting the sine terms equal,

15 $(-\sigma_{\text{H}} + \sigma_{\text{H}}')k_x + (\sigma_{\text{P}} + \sigma_{\text{P}}')k_y = E_0(-\sigma_{\text{P}} + \sigma_{\text{P}}')$ (18)

These two linear equations in $k_x$ and $k_y$ are

$$
\begin{bmatrix} \sigma_{\text{P}} + \sigma_{\text{P}}' & \sigma_{\text{H}} - \sigma_{\text{H}}' \\ -\sigma_{\text{H}} + \sigma_{\text{H}}' & \sigma_{\text{P}} + \sigma_{\text{P}}' \end{bmatrix}\begin{bmatrix} k_x \\ k_y \end{bmatrix} = E_0\begin{bmatrix} \sigma_{\text{H}} - \sigma_{\text{H}}' \\ -\sigma_{\text{P}} + \sigma_{\text{P}}' \end{bmatrix}
\tag{19}
$$

$$
\left[[\sigma]' + [\sigma]^T\right]\begin{bmatrix} k_x \\ k_y \end{bmatrix} = \left[[\sigma]' - [\sigma]\right]\begin{bmatrix} 0 \\ E_0 \end{bmatrix}
\tag{20}
$$

20 The LHS represents the rate of loss of net charge from the dipole $\sigma = 2\varepsilon_0 E_{\text{dip}}$. On the RHS, the *difference* in $[\sigma]$ between the patch and its surroundings is forcing its polarisation. Hence we can write

$$
S\boldsymbol{E}_{\text{dip}} = (n-1)[\sigma]\boldsymbol{E}_0
\tag{21}
$$

where the matrix on the LHS is

$$S = \begin{bmatrix} \sigma_P(n+1) & \sigma_H(-n+1) \\ \sigma_H(n-1) & \sigma_P(n+1) \end{bmatrix} \tag{22}$$

which is *nearly* scalar for $n \approx 1$, but let us define $H$ by writing

$$S = (n+1)\sigma_P \begin{bmatrix} 1 & -\eta\sigma_H/\sigma_P \\ \eta\sigma_H/\sigma_P & 1 \end{bmatrix} = (n+1)\sigma_P H \tag{23}$$

$$(n+1)\sigma_P H \boldsymbol{E}_{\text{dip}} = (n-1)[\sigma]\boldsymbol{E}_0 \tag{24}$$

Solving for $\boldsymbol{E}_{\text{dip}}$, a polarisation with

$$\boldsymbol{E}_{\text{dip}} = \frac{\eta}{\sigma_P} H^{-1}[\sigma]\boldsymbol{E}_0 \tag{25}$$

yields a divergence-free current field.

The steady-state electric field in and around the patch, using Eqs. (A7) and (A8), is

$$\boldsymbol{E}_{\text{int}} = \left( I - \frac{\eta}{\sigma_P} H^{-1}[\sigma] \right) \boldsymbol{E}_0 \tag{26}$$

$$\boldsymbol{E}_{\text{ext}} = \left( I + \frac{\eta R^2}{\sigma_P \rho^2} D(\theta) H^{-1}[\sigma] \right) \boldsymbol{E}_0 \tag{27}$$

From this expression for $\boldsymbol{E}_{\text{int}}$, one can verify that it is at an angle relative to $\boldsymbol{E}_0$ whose tangent is $\eta\sigma_H/\sigma_P$. This agrees with Hysell and Drexler's Eq. (9), which gives us confidence in our results, although we focus below on the *boundary*'s drift, rather than that of the ions inside.

## 3.2 Simplifying assumptions

We shall deal here with a single ion species, and assume that electrons are fully magnetised. These assumptions are not necessary for a unique solution, but will greatly simplify the algebra. Under these assumptions, and using the ion magnetisation parameter $\kappa_i = \omega_i/\nu_{\text{in}}$, we have

$$\sigma_P = \frac{q_i n_i}{B} \left( \frac{\kappa_i}{1+\kappa_i^2} \right) \tag{28}$$

$$\sigma_H = \frac{q_i n_i}{B} \left( \frac{1}{1+\kappa_i^2} \right) \tag{29}$$

Thus $\sigma_P = \kappa_i \sigma_H$, and $|\kappa_e|$ is much larger than both $\kappa_i$ and unity.

We also assume no neutral drift, or equivalently that the electric field and all of the species' drifts are expressed in the neutrals' frame of reference. Finally, we assume a vertical magnetic field, but this is addressed in the Discussion.

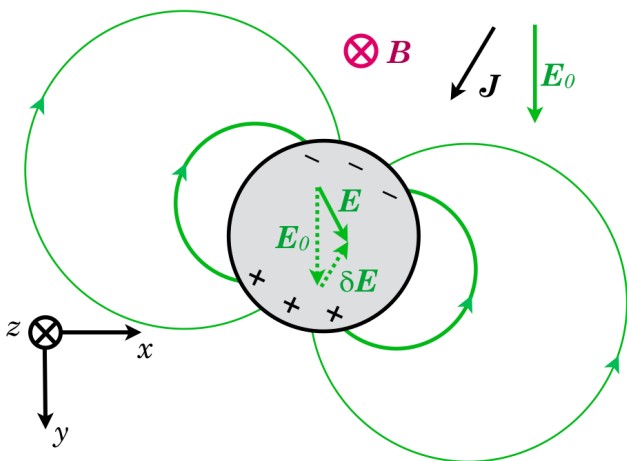

**Figure 4.** The Hall current, not yet shown in Fig. 2, adds a rotation to the polarisation of the patch, however the exact solution for a sharp circular boundary is still a cylindrical dipole.

### 3.3 Species drift

The drift of a species $s$ (ion or electron) in the plane perpendicular to $\boldsymbol{B}$ can be described by

$$\mathbf{v}_s = \begin{bmatrix} \mu_\mathrm{P} & \mu_\mathrm{a} \\ -\mu_\mathrm{a} & \mu_\mathrm{P} \end{bmatrix}_s \boldsymbol{E} \tag{30}$$

where

$$\mu_\mathrm{a} = \frac{\kappa_s^2}{B(1+\kappa_s^2)} \text{ and } \mu_\mathrm{P} = \frac{\kappa_s}{B(1+\kappa_s^2)} \tag{31}$$

are the ambipolar and Pedersen mobilities, respectively. If we let $\phi_s$ be an angle defined by $\kappa_s = \cot(\phi_s)$ then we can write the mobility matrix as

$$\begin{aligned} \mathbf{v}_s &= \frac{1}{B} \begin{bmatrix} \sin\phi\cos\phi & \cos^2\phi \\ -\cos^2\phi & \sin\phi\cos\phi \end{bmatrix}_s \boldsymbol{E} \\ &= \frac{\cos\phi_s}{B} \begin{bmatrix} \sin\phi & \cos\phi \\ -\cos\phi & \sin\phi \end{bmatrix}_s \boldsymbol{E} \\ &= \frac{\cos\phi_s}{B} R(\phi_s - \tfrac{\pi}{2}) \boldsymbol{E} \end{aligned} \tag{32}$$

where $R$ is a rotation matrix, anticlockwise if looking from positive $z$. We use $R_s$ below as shorthand for $R(\phi_s - \frac{\pi}{2})$. Alternately, since $\mathbf{v}_a = B^{-1} R(-\frac{\pi}{2}) \boldsymbol{E}$,

$$
\mathbf{v}_s = \cos\phi_s \begin{bmatrix} \cos\phi & -\sin\phi \\ \sin\phi & \cos\phi \end{bmatrix}_s \mathbf{v}_a
$$

$$
= \cos\phi_s R(\phi_s) \mathbf{v}_a \tag{33}
$$

where $\mathbf{v}_a$ is the ambipolar drift. We see that $\phi_s$ is the angle which the species' drift makes w.r.t. the ambipolar drift, however we shall use the previous expression in order to get $\mathbf{v}_s$ as a function of $\boldsymbol{E}$.

### 3.4 Patch drift speed

Applying Eq. (4) to our dipole field,

$$
\hat{\boldsymbol{n}} \cdot \mathbf{v}_b = \frac{\cos\phi_s}{(n-1)B} \, \hat{\boldsymbol{n}} \cdot (n R_s \boldsymbol{E}_{\text{int}} - R_s \boldsymbol{E}_{\text{ext}})
$$

$$
= \frac{\cos\phi_s}{(n-1)B} \, \hat{\boldsymbol{n}} \cdot R_s \left[ n \left( I - \frac{\eta}{\sigma_P} H^{-1}[\sigma] \right) - \left( I + \frac{\eta}{\sigma_P} D H^{-1}[\sigma] \right) \right] \boldsymbol{E}_0
$$

$$
= \frac{\cos\phi_s}{(n-1)B} \, \hat{\boldsymbol{n}} \cdot R_s \left[ (n-1)I - \frac{\eta}{\sigma_P}(nI + D) H^{-1}[\sigma] \right] \boldsymbol{E}_0 \tag{34}
$$

We now make use of Eq. (A11), the fact that $\hat{\boldsymbol{n}}^T R_s D = \hat{\boldsymbol{n}}^T R_s^T$. It can also be shown that

$$
n R_s + R_s^T = (n+1)\sin\phi_s \begin{bmatrix} 1 & \eta\kappa_s \\ -\eta\kappa_s & 1 \end{bmatrix} \tag{35}
$$

Let us call that last matrix $M_s$. Simplifying,

$$
\hat{\boldsymbol{n}} \cdot \mathbf{v}_b = \hat{\boldsymbol{n}} \cdot \left[ \frac{\cos\phi_s}{B} R_s \boldsymbol{E}_0 - \frac{\sin\phi_s \cos\phi_s}{B\sigma_P} M_s H^{-1}[\sigma] \boldsymbol{E}_0 \right] \tag{36}
$$

The factor in square brackets on the RHS of the last equation has no $\theta$ dependence, and the tangential component of $\mathbf{v}_b$ is arbitrary. So we may drop the $\hat{\boldsymbol{n}}$ from both sides and choose

$$
\mathbf{v}_b = \frac{\cos\phi_s}{B} R_s \boldsymbol{E}_0 - \frac{\sin\phi_s \cos\phi_s}{B\sigma_P} M_s H^{-1}[\sigma] \boldsymbol{E}_0
$$

$$
= \mathbf{v}_{s,0} - \frac{\sin\phi_s \cos\phi_s}{B\sigma_P} M_s H^{-1} \boldsymbol{J}_0 \tag{37}
$$

where the first term on the RHS is the background drift of the species, and we have used $\boldsymbol{J} = [\sigma]\boldsymbol{E}$. We now use $\boldsymbol{J} = q_i n_i (\mathbf{v}_i - \mathbf{v}_e)$; $\sigma_P = q_i n_i \kappa_i / B(1 + \kappa_i^2)$; and $\kappa_i = \cot\phi_i$ to get (dropping the subscript 0 now since all quantities except $\eta$ are background values)

$$\mathbf{v}_b = \mathbf{v}_s - \frac{\sin\phi_s\cos\phi_s}{\sin\phi_i\cos\phi_i} M_s H^{-1}(\mathbf{v}_i - \mathbf{v}_e)$$

$$= \mathbf{v}_s + \frac{\sin\phi_s\cos\phi_s}{\sin\phi_i\cos\phi_i}
\begin{bmatrix} 1 & \eta\kappa_s \\ -\eta\kappa_s & 1 \end{bmatrix}
\begin{bmatrix} 1 & -\eta/\kappa_i \\ \eta/\kappa_i & 1 \end{bmatrix}^{-1}
(\mathbf{v}_e - \mathbf{v}_i)$$

$$= \mathbf{v}_s + \frac{\sin\phi_s\cos\phi_s}{\sin\phi_i\cos\phi_i}
\begin{bmatrix} 1 & \eta\kappa_s \\ -\eta\kappa_s & 1 \end{bmatrix}
\frac{\kappa_i}{\kappa_i^2+\eta^2}
\begin{bmatrix} \kappa_i & \eta \\ -\eta & \kappa_i \end{bmatrix}
(\mathbf{v}_e - \mathbf{v}_i)$$

$$= \mathbf{v}_s + \frac{\sin\phi_s\cos\phi_s}{\sin^2\phi_i(\kappa_i^2+\eta^2)}
\begin{bmatrix} 1 & \eta\kappa_s \\ -\eta\kappa_s & 1 \end{bmatrix}
\begin{bmatrix} \kappa_i & \eta \\ -\eta & \kappa_i \end{bmatrix}
(\mathbf{v}_e - \mathbf{v}_i) \tag{38}$$

This equation ought to yield the same answer for either ions or electrons. We first demonstrate this for $|\eta| \ll 1$.

For ions ($s = i$) we obtain

$$\mathbf{v}_b = \mathbf{v}_i + \frac{\cot\phi_i}{\kappa_i^2+\eta^2}
\begin{bmatrix} \kappa_i(1-\eta^2) & \eta(\kappa_i^2+1) \\ -\eta(\kappa_i^2+1) & \kappa_i(1-\eta^2) \end{bmatrix}
(\mathbf{v}_e - \mathbf{v}_i)$$

$$= \mathbf{v}_i + \begin{bmatrix} 1 & \eta(\kappa_i^2+1)/\kappa_i \\ -\eta(\kappa_i^2+1)/\kappa_i & 1 \end{bmatrix}
(\mathbf{v}_e - \mathbf{v}_i) + O(\eta^2) \tag{39}$$

The matrix is approximately a rotation, anticlockwise if we are looking parallel to $B$, by a small angle

$$\alpha \approx \frac{\eta(\kappa_i^2+1)}{\kappa_i}$$

$$= \frac{(n-1)\csc^2\phi_i}{(n+1)\cot\phi_i}$$

$$\approx \frac{(n-1)}{\sin 2\phi_i} \tag{40}$$

where in the last step we use $n + 1 \approx 2$. See Fig. 5 for a sketch of this construction of $\mathbf{v}_b$.

For electrons ($s = e$) we obtain, using $\kappa_e \ll -1$

$$\mathbf{v}_b = \mathbf{v}_e + \frac{\sin\phi_e\cos\phi_e}{\sin^2\phi_i(\kappa_i^2+\eta^2)}
\begin{bmatrix} 1 & \eta\kappa_e \\ -\eta\kappa_e & 1 \end{bmatrix}
\begin{bmatrix} \kappa_i & \eta \\ -\eta & \kappa_i \end{bmatrix}
(\mathbf{v}_e - \mathbf{v}_i)$$

$$= \mathbf{v}_e + \frac{\sin\phi_e\cos\phi_e}{\sin^2\phi_i\cot^2\phi_i}\eta\kappa_e
\begin{bmatrix} 0 & 1 \\ -1 & 0 \end{bmatrix}
\kappa_i
\begin{bmatrix} 1 & 0 \\ 0 & 1 \end{bmatrix}
(\mathbf{v}_e - \mathbf{v}_i) + O(\eta^2)$$

$$= \mathbf{v}_e + \eta\frac{\cos^2\phi_e}{\sin\phi_i\cos\phi_i}R(-\tfrac{\pi}{2})(\mathbf{v}_e - \mathbf{v}_i)$$

$$= \mathbf{v}_e + \frac{(n-1)}{\sin 2\phi_i}R(-\tfrac{\pi}{2})(\mathbf{v}_e - \mathbf{v}_i) + O(\kappa_e^{-1}) \tag{41}$$

Fig. 5 also sketches this construction of $\mathbf{v}_b$, and one can see it is coïncident with the value of $\mathbf{v}_b$ obtained using the ions.

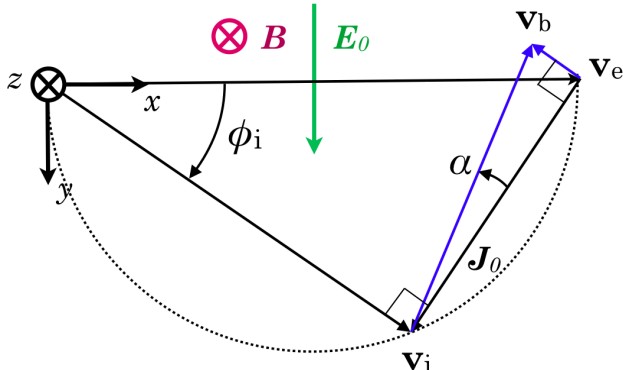

**Figure 5.** The dashed semicircle shows the locus of $\mathbf{v}_i$ for various values of $\kappa_i$. The blue arrows show the construction of $\mathbf{v}_b$, the drift of the density enhancement, relative to either the ion or electron background drift, for a modest enhancement ($n \gtrsim 1$).

The magnitude of this drift, relative to the electron ($\approx$ ambipolar) drift is

$$
\begin{aligned}
v_b &= \frac{(n-1)}{\sin 2\phi_i} |\mathbf{v}_e - \mathbf{v}_i| \\
&= \frac{(n-1)}{\sin 2\phi_i} \sin\phi_i v_a \\
&\approx \tfrac{1}{2}(n-1)\sec\phi_i v_a
\end{aligned}
\tag{42}
$$

5 and the orientation of the patch's drift is at right angles to $\boldsymbol{J}_0$. It is slower than $v_a$ for the case of density enhancements and faster for density depletions. Curiously, the drift of an enhancement's boundary has a component *against* the electric field.

### 3.5  General solution for arbitrary relative density

Still restricting ourselves to the three simplifying assumptions (one ion species, magnetised electrons, no neutral drift) we can determine the drift velocity of a patch or depletion with $n$ much smaller or larger than unity.

10 Using Eq. (38), with either species, we can show that in the limit of large $n$, $\mathbf{v}_b \to 0$. In the limit of $n \to 0$ (a deep hole), we get $\mathbf{v}_b \to 2\mathbf{v}_i$. And of course for $n=1$ we have $\mathbf{v}_b = \mathbf{v}_a$. Looking at the trend for $n \sim 1$ suggested by Fig. 5, and using the intuition obtained with a bit of numerical experimentation, one can appreciate that the range of values which $\mathbf{v}_b$ can take on as a function of $n$, for a fixed value of $\kappa_i$, describes a circular arc passing through the origin, $\mathbf{v}_a$ and $2\mathbf{v}_i$.

Since the origin and $\mathbf{v}_a$ both lie on the circular arc, the arc's centre must lie on the line $v_x = \tfrac{1}{2}v_a$. Furthermore, the trend 15 around $n \sim 1$ being perpendicular to $\boldsymbol{J}$ shows that the centre lies along the line of $\boldsymbol{J}$ extended from $\mathbf{v}_a$. Thus the centre is at $\mathbf{v}_{\text{cent}} = \tfrac{1}{2}v_a[1, \cot\phi_i]^T$. So we can separate the centre from the rest of Eq. (38) and, using $s = e$ and repeating the same $O(\kappa_e^{-1})$

approximations used to get Eq. (41), write

$$\mathbf{v}_b = \mathbf{v}_{\text{cent}} - \frac{1}{2} \begin{bmatrix} 1 \\ \cot\phi_i \end{bmatrix} v_a + \mathbf{v}_e + \frac{\eta}{\sin^2\phi_i(\kappa_i^2 + \eta^2)} \begin{bmatrix} -\eta & \kappa_i \\ -\kappa_i & -\eta \end{bmatrix} (\mathbf{v}_e - \mathbf{v}_i)$$

$$= \mathbf{v}_{\text{cent}} + \frac{1}{2} \begin{bmatrix} 1 \\ -\cot\phi_i \end{bmatrix} v_a + \frac{\csc^2\phi_i\,\eta}{(\kappa_i^2 + \eta^2)} \begin{bmatrix} -\eta & \kappa_i \\ -\kappa_i & -\eta \end{bmatrix} \sin\phi_i \begin{bmatrix} \sin\phi_i \\ -\cos\phi_i \end{bmatrix} v_a$$

$$= \mathbf{v}_{\text{cent}} + \frac{1}{2}\csc\phi_i R(\phi_i - \tfrac{\pi}{2}) \begin{bmatrix} 1 \\ 0 \end{bmatrix} v_a + \frac{\csc\phi_i\,\eta}{(\kappa_i^2 + \eta^2)} \begin{bmatrix} -\eta & \kappa_i \\ -\kappa_i & -\eta \end{bmatrix} R(\phi_i - \tfrac{\pi}{2}) \begin{bmatrix} 1 \\ 0 \end{bmatrix} v_a$$

$$= \mathbf{v}_{\text{cent}} + \frac{\csc\phi_i}{2(\kappa_i^2 + \eta^2)} R(\phi_i - \tfrac{\pi}{2}) \left( (\kappa_i^2 + \eta^2) \begin{bmatrix} 1 \\ 0 \end{bmatrix} + 2\eta \begin{bmatrix} -\eta \\ -\kappa_i \end{bmatrix} \right) v_a$$

$$= \mathbf{v}_{\text{cent}} + \frac{\csc\phi_i}{2(\kappa_i^2 + \eta^2)} R(\phi_i - \tfrac{\pi}{2}) \begin{bmatrix} \kappa_i^2 - \eta^2 \\ -2\kappa_i\eta \end{bmatrix} v_a \tag{43}$$

Let $\beta$ be an angle defined by $\tan\beta = \eta/\kappa_i$ (this is the angle between $\boldsymbol{E}_{\text{int}}$ and $\boldsymbol{E}_0$). Then

$$\mathbf{v}_b = \mathbf{v}_{\text{cent}} + \frac{\csc\phi_i}{2(1 + \tan^2\beta)} R(\phi_i - \tfrac{\pi}{2}) \begin{bmatrix} 1 - \tan^2\beta \\ -2\tan\beta \end{bmatrix} v_a$$

$$= \mathbf{v}_{\text{cent}} + \frac{1}{2}\csc\phi_i R(\phi_i - \tfrac{\pi}{2}) \begin{bmatrix} \cos 2\beta \\ \sin 2\beta \end{bmatrix} v_a$$

$$= \mathbf{v}_{\text{cent}} + \frac{1}{2}\csc\phi_i R(\phi_i - 2\beta - \tfrac{\pi}{2}) \mathbf{v}_a \tag{44}$$

Fig. 6 illustrates the portion of the arc along which the boundary drift $\mathbf{v}_b$ can occur for various density ratios $n$, for one particular value of $\kappa_i$.

If we look at the electron drift velocity inside the patch using Eqs. (26) and (32), we find that the electrons drift along with the boundary. So the patch (or depletion) keeps its original electrons (so to speak) whereas the ions are transiently within it, as in the slab geometry.

In Fig. 7 we show a series of possibilities for different ion magnetisation ratios from 0.1 to 10. Each coloured arc shows, for a fixed value of $\kappa_i$, how patches or depletions (varying $n$, or $\eta$) should drift. An enhancement always drifts slower than ambipolar, with a component against $\boldsymbol{E}$, whereas a depletion always has a component parallel to $\boldsymbol{E}$. Usually a depletion drifts faster than ambipolar, but with highly demagnetised ions an enhancement's drift can be slower.

In the frame of reference of the background drift, the current vector $\boldsymbol{J}$ separates the two cases, with enhancements drifting towards its right side (boreal pole), and depletions towards its left.

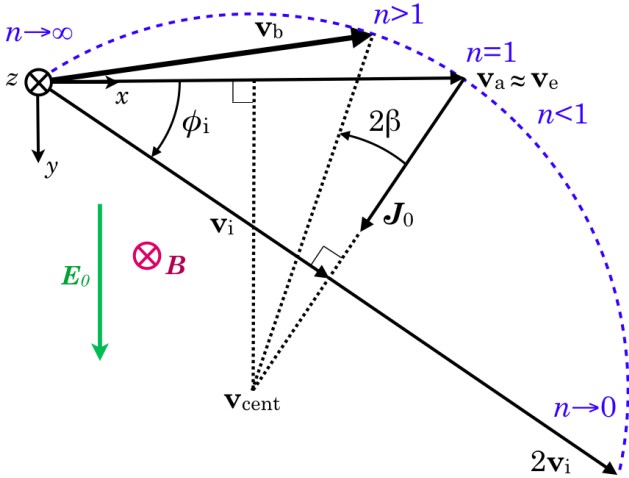

**Figure 6.** The construction of the circular arc (blue dotted curve) along which the drift of the boundary of a circular patch ($\mathbf{v}_b$) can occur for a given ion magnetisation $\kappa_i$. The blue text shows the variation of the drift along the arc for various values of the relative density $n$, which determines the angle $\beta$. In this example $\kappa_i \sim 1.5$.

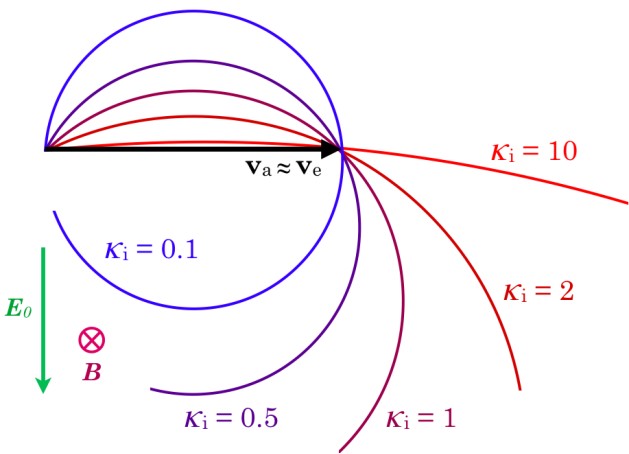

**Figure 7.** The variation w.r.t. ion magnetisation $\kappa_i$ of the arc along which the drift of a circular patch can lie. All of the arcs coïncide at the origin (the limit of high $n$) and at the ambipolar drift (for the case of $n=1$). An enhancement always drifts slower than ambipolar, with a component against $\mathbf{E}$, whereas a depletion always has a component parallel to $\mathbf{E}$.

## 4  Discussion

We have derived the drift behaviour of a circular density enhancement (or depletion) in a 2-D magnetised plasma. Our study grew from a 2-D auroral modeling effort in the *meridional* plane (de Boer et al., 2010) which led to thinking about appropriate upper boundary conditions ($\sim 1000$ km) for the electric potential, and the time scales over which various effects should assert

themselves. The eventual conclusion was that $J_\parallel = 0$ was appropriate over the time scales of interest, and this inquiry led to the arguments presented above in justifying the 2-D analysis in a surface *perpendicular to B*.

The idea that charge accumulation from a non-uniform conductivity $[\sigma]$ would set up an electric field disturbance in a way that would force the system towards steady-state $(\nabla \cdot J = 0)$ is simple, although the algebra turned out to be more complicated than expected. Perhaps a shorter derivation is possible. But we found a unique result that we are confident in.

Still, one might challenge the relevance of our result in the limit as $\kappa_i \to \infty$, since the time required to approach an equilibrium condition also tends to infinity. However by following other lines of argument not included in this paper, and considering the effective, perpendicular *permittivity* of a magnetised plasma with $\kappa_i \to \infty$, it can be shown that the steady-state towards which the E-region is driving the convection pattern is the same as the $E_\perp$ structure obtained from considering a circular patch (or depletion) with $E$ initially zero, and increasing $E_0$ up to some steady value.

One should also reasonably question how these 2-D results – specifically Eq. (44) – can be extended to the real ionosphere, where $\kappa_i$ and $\phi_i$ vary continuously with height. Each altitude layer will be forcing a different patch drift speed, while parallel conductivity is trying to enforce a coherent flux-tube drift. A conjecture (which awaits analytic justification) is that the *effective* ion magnetisation of a flux tube can be obtained from the ratio of height-integrated conductivities:

$$\kappa_{\text{eff.}} = \Sigma_P / \Sigma_H \tag{45}$$

and that values of $\phi_i$ and $\beta$ determined from this $\kappa_{\text{eff.}}$ can be used in Eq. (44) to find the theoretical drift direction and speed of a patch (or a depletion). Of course such a 3-D drift structure would entail dipolar structures of FACs closed *within the ionosphere*. There would be a special height at which $\kappa_{\text{eff.}} = \kappa_i$, and the dipolar FAC would be oppositely oriented above and below this height.

## 4.1 Addressing some idealisations

If we are considering F-region structure, our results are still applicable. We presented the analysis in the context of the E-region because we are treating the Hall current, whereas the F-region alone, with negligible Hall current, would constitute a narrower problem. Also any E-region structure is weighted much more highly than the F-region's in determining electric field structure, due to its stronger contribution to $\Sigma_P$.

We assumed a vertical $B$ field. In the polar region this is not a large approximation, but the $E \times B$ drift acts perpendicular to $B$ so it can have a vertical component. In the polar cap, between the dayside open-close boundary (OCB) and the line across 06-18 MLT, the convection adds an upward component to the plasma's vertical momentum balance, while on the midnight side of the cap it is driving the plasma downwards. This adds a layer of complexity, but also ensures that even initially purely F-region patches should create some conductivity structure by the time the plasma reaches the nightside OCB.

It may be possible to extend our analysis to elliptical patches. However it has some complexities that might only yield to complex analysis, as Hysell and Drexler (2006) have accomplished. At least this study reinforces their result for circular patches.

We have studied patches with sharp, step boundaries in density, which is an idealisation. The warm-plasma mechanism of ion diffusion, which operates on time and distance scales not addressed in this paper, would gradually degrade such a sharp step, as would any instabilities along the boundary, for example the shear-driven instability (SDI). However one can also see that any perturbation of the boundary, while suffering some SDI growth at the 12 and 6 o'clock positions in Fig. 3, will ultimately be convected towards the 3 o'clock position, where the shear approaches zero. This would prevent the growth of longer-wavelength (hence slower-growing) modes more than short-wavelength (faster-growing) ones. So SDI would also gradually blunt the sharpness of the boundary, but it does not necessarily imply the complete breakup of the patch as a distinct entity. We examine the sharp, circular case because it yields to analysis, and because we can make useful qualitative arguments based on the results.

There is an open question which requires further research: how non-ideal is the electric potential source along a given flux tube? Our research began with modelling closed field lines in the dawn- or dusk-side auroral oval, and these are in a situation which is most directly addressed in our Introduction and in App. B. A Neumann upper boundary condition is more appropriate for electric potential. An old open flux tube in the cap, especially as it approaches the nightside OCB, should also display density-driven potential structuring. A new open field line will behave the most like an ideal power source, i.e. one which can provide currents as required to maintain the potential that is initially mapped down. Here a Dirichlet boundary condition may be more appropriate. However the opportunity to observe and measure also progresses in the same sequence, with dayside polar cap patches being the largest and most prominent phenomena available for quantitative study. So our discussion in Sec. 4.3 will necessarily focus on a region where the effect we have put forth may not be dominant.

### 4.2 Implications for gradient scale lengths

It is impossible to extend these analytical methods to arbitrary shapes. But it seems clear that a patch with concentric contours of density would have a progressive drift, relative to ambipolar, that grows as one looks deeper within the patch. As a thought experiment, consider a small, denser patch within a larger patch. While an analytical result may be elusive except for the moment at which they are concentric, and neither will remain exactly circular, qualitatively we know that the inner patch will drift within the larger one, and that this relative drift will see it approach the outer boundary on the side opposite to $\mathbf{v}_i$. (The curves in Fig. 7 are tangent to $\mathbf{v}_i$ at $\mathbf{v}_a$.) Fig. 8 shows for one case how this might appear.

Therefore we expect to see that any gradual density features would evolve in such a way that their gradients 'pile up' on one side and get stretched out on the other. In the case of banded structures, this may produce something like a saw-tooth density profile. If $\nabla_\perp \rho_m$ is the gradient of plasma mass density $\rho_m$, then the steepest gradients should be found parallel to the *ion* drift (not the $\boldsymbol{E} \times \boldsymbol{B}$ drift), with $\nabla_\perp \rho_m$ oriented in the same sense.

Such a steepening of mass density gradients would entail polarisations and electric field structures on the same scale lengths. Unless these are already uniform along magnetic field lines, FACs will arise so that the plasma along any given flux tube accelerates to regain a coherent drift. One of the conclusions of de Boer et al. (2010) was that the ionosphere's response to precipitation could be FAC over a much *larger* region than the precipitation, and that FAC structure was a convolution of the precipitation structure, so that it always had larger gradient scale lengths. However in this paper we see the possibility of the

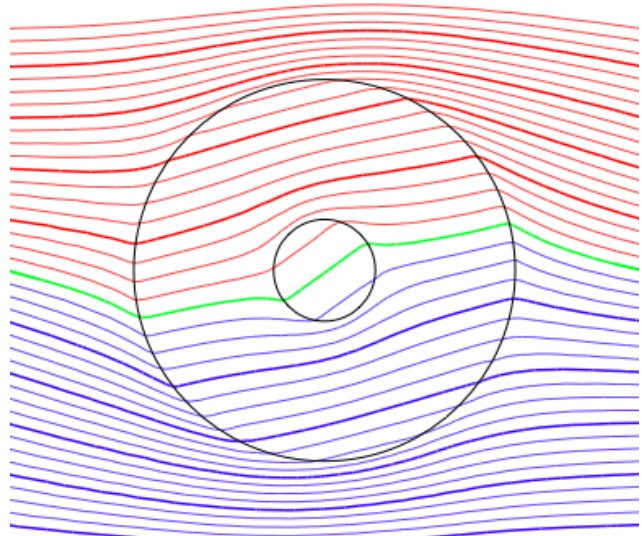

**Figure 8.** Equipotential contours that satisfy $\nabla \cdot \boldsymbol{J} = 0$ for two concentric patches of density $2\times$ and $5\times$ the background density, and a uniform $\kappa = 1$. Although the $\boldsymbol{E}$ field and the boundary shapes will become quite complicated as the inner patch approaches the boundary of the outer one, we can appreciate that the step boundaries will approach each other around their 10 o'clock position, while growing farther apart around 4 o'clock.

ionosphere developing very short gradient scale lengths and generating FAC over equally small scales, *smaller than any initial density structure*, without the requirement for precipitation to initiate the fine-scale structure.

This effect could explain why in auroral phenomena we see structures cascading to smaller and smaller gradient scale lengths. The initial density gradients generated in the auroral oval by precipitation are stronger to begin with than at other
latitudes. Also the strong electric fields found there reduce the time scale for features to cascade to smaller spatial scales. This combination of conditions yields an increased chance that this cascade time scale might prevail over the erasure of structure on the scale of the ion chemical lifetime.

### 4.3    Search for observational support

The predicted structuring of the electric field around density features, and the relative drift of those features, is independent
of scale. Hysell and Drexler were motivated by Farley-Buneman waves. We began with the goal of understanding small-scale auroral structure, however an observational test of our results for individual features would appear to be very difficult. The most promising avenue for testing our hypotheses will be in the observation of large-scale patches in the polar cap and auroral oval. Both Carlson (2012) and Zhang et al. (2013) have provided analyses of events using, respectively, EISCAT for Ne data and GPS receiver arrays for TEC. The latter dataset is overlaid with SuperDARN convection patterns. These papers show
incontrovertibly that patches of ionisation do cross the polar cap, and can return in the sunward return flow. However within

the spatial and temporal resolution of their available data, it is challenging to see either confirmation or refutation of the electric-field structuring that we posit.

There are however two ways that the effect we argue could bias patch motion statistically. Fig. 7 shows that a patch in the boreal polar cap *might* be expected to drift somewhat to the left (towards post-midnight) relative to the average convection, and that it should travel more *slowly* than the mean drift. But if it is a predominantly F-region feature, then the leftward deviation may be negligible.

Moen et al. (2007) did a statistical study of patch exit times and found a high degree of symmetry around midnight. There was a small bias toward pre-midnight exits, the distribution being centred on 23:25 MLT, which might only reflect the small bias of IMF By in their sample. This offset is the opposite of what we are positing, albeit in a region where convection is expected to be the least structured by conductivity. Moen et al. (2015) examined the statistical variation of patch exit times with IMF By and Bz more closely, and found that the pre-midnight shift for By positive is mirrored by a post-midnight shift for By negative. So this test is inconclusive.

Next we look at drift speed. Oksavik et al. (2010) studied two particular polar cap patch events in 2001, of which we examine no. 2. Their Fig. 3.e) is interesting because it shows that this patch's velocity was distinctly lower than the plasma ahead and behind it. The SuperDARN flow plotted in their Fig. 2.d) shows a speed of approximately 470 m/s on the patch's right (east) side and 620 m/s on its left. (The patch undergoes a clockwise rotation of $90°$.) The SuperDARN *maps* for that day show electric field strengths of about 40 and 30 mV/m at 08 and 09 UT, respectively, or roughly 700 m/s at 0830. Yet the patch progressed through the east and west beams of the radar at 226 and 566 m/s, respectively. So there is some evidence that patches convect more slowly than the average convective flow around them.

Hosokawa et al. (2010) looked at an event where two patches were pulled away from each other by a shear in the convection. It is curious that in their Fig. 6 the highest speed shown is inferred from the SuperDARN convection map, and is clearly higher than the patch's speed measured by radar backscatter. Also Gillies et al. (2009) have examined the factor of about 0.75 by which SuperDARN Doppler velocities are lower than those obtained for convection from the DMSP satellites. Their work shows that about a third of the discrepancy is accounted for by taking the index of refraction into account. But it is intriguing to speculate that the remaining factor may arise because the DMSP data yield *mean* convection, while the Doppler speeds may be biased towards plasma with higher density and therefore stronger radar return. Perhaps the remaining systematic difference is due to an inverse correlation between plasma density and *local* electric field strength.

In some convection maps, it seems like it might even be possible to see the effect we posit. For example in Fig. 2 in Zhang et al., looking around the terminator in the polar cap in panels C, D & E, and in the return flow in panel G, it appears that the contours of potential are slightly spread out (weaker $E$) around the stronger TEC structures.

## 4.4 Modeling

We should address how our work is relevant to ionospheric modeling. Our result shows that the electric field cannot be simply prescribed for some region, but that it will have structure implicitly determined by the plasma density structure. Most models,

including our own cited work, assume an $\boldsymbol{E}$ or potential field that is prescribed in some way. For example Schunk and Sojka (1987) used an $\boldsymbol{E}$ field that remained fixed despite the introduction of very strong density features.

A numerical model intended to address $\boldsymbol{E}$ field structuring might begin with an initial electric potential map, but the actual Pedersen and Hall currents will generate FACs wherever they converge or diverge in the ionosphere. These FACs cannot be driven immediately or indefinitely by magnetospheric processes. Charge accumulations will then force a structuring of the ionospheric and magnetospheric potential towards a situation where FACs are no longer required to maintain current closure, i.e. exactly the sort of structure we have identified. Where the plasma density is higher, the field will be lower, and *vice versa*. In the limit of closed field lines, this will amount to solving the Laplace equation in 2-D.

## 5  Conclusions

The following characteristics of $\boldsymbol{E} \times \boldsymbol{B}$ drift in 2-D, magnetised plasma, shown by Hysell and Drexler (2006), have been confirmed through an alternate analysis:

1. While plasma on an open flux tube may have a uniform electric field more-or-less "imposed" on it regardless of density structure, plasma on closed flux tubes will experience a structuring of the steady-state electric field that depends on density features – weighted towards dependence on E-region density.

2. For a circular density feature, the assumption of a dipolar net charge with appropriate magnitude and orientation can yield a divergence-free current field.

3. A density feature does not "own" a particular parcel of ions – the ions both inside and out can convect through the boundary – nevertheless the boundary of a circular density feature retains a circular shape, and the electrons convect with the density feature.

We have also shown that:

4. The boundary of a circular feature should convect with a velocity given by Eq. (44) and shown in Fig. 7 – always slower than ambipolar for an enhancement and usually faster for a depletion; and with a component against or with, respectively, the background electric field.

5. An obvious extension of the result for a sharp, circular feature is that features with density *gradients* will see gradients on one side steepened and gradients on the other side weakened.

6. The E-region can therefore generate smaller-scale structure than its initial structure, at any length scale and even without instability present, and the time scale for this to occur is inversely related to the electric field strength.

As well, we have provided some arguments as to why and how these 2-D results are still applicable to the real ionosphere with its altitude-dependence of plasma properties. Moreover, we wish to show in a future paper that the structuring described at point 1., in a *non-conducting* plasma, will depend on plasma *mass* density features.

We wish to remind the reader that we present only a 2-D analysis, which is effectively a *steady-state* picture where the FACs that must be generated, at least transiently, by our model's predicted convection have had time to propagate and to impress that convection pattern back onto the small-scale convection of the magnetosphere. Our Discussion and App. C both attempt to address this limitation in the application of our model. But we hope that this model will generate discussion and facilitate a more realistic and complete model.

*Code availability.* n/a

*Data availability.* n/a

*Code and data availability.* n/a

## Appendix A: Cylindrical dipole

Let a circle of radius $R$ have a surface charge density $\sigma = s \cos\theta$. The electric field inside is uniform: $\boldsymbol{E}_{\text{int}} = -a\hat{\boldsymbol{x}}$. Outside it has the form $E_\rho = b\rho^{-2}\cos\theta$ and $E_\theta = b\rho^{-2}\sin\theta$.

At the centre, using an expression for the electric field around an infinite line of charge and symmetry,

$$a = |\boldsymbol{E}_{\text{int}}| = 2 \int\limits_{-\pi/2}^{\pi/2} \frac{sR\cos\theta}{2\pi\varepsilon_0 R} \cos\theta \, d\theta$$

$$= \frac{2s}{2\pi\varepsilon_0} \cdot 2 \int\limits_0^{\pi/2} \cos^2\theta \, d\theta$$

$$= \frac{2s}{\pi\varepsilon_0} \cdot \frac{\pi}{4} = \frac{s}{2\varepsilon_0} \tag{A1}$$

At the pole ($\theta = 0$), using Gauss' law,

$$\boldsymbol{E}_{\text{ext}} - \boldsymbol{E}_{\text{int}} = \frac{s}{\varepsilon_0}\hat{\boldsymbol{x}} \tag{A2}$$

$$bR^{-2} + \frac{s}{2\varepsilon_0} = \frac{s}{\varepsilon_0} \tag{A3}$$

$$b = \frac{sR^2}{2\varepsilon_0} = aR^2 \tag{A4}$$

The polarisation vector $\boldsymbol{P}$ inside the circle is $s\hat{\boldsymbol{x}}$. The field outside is

$$
\begin{aligned}
\boldsymbol{E}_{\text{ext}} &= \frac{b}{\rho^2}(\cos\theta\,\hat{\boldsymbol{\rho}} + \sin\theta\,\hat{\boldsymbol{\theta}}) \\
&= \frac{sR^2}{2\varepsilon_0\rho^2}\begin{bmatrix} 2\cos^2\theta - 1 \\ 2\sin\theta\cos\theta \end{bmatrix}
\end{aligned}
\tag{A5}
$$

If we generalise the charge dipole to an arbitrary orientation:

$\sigma = s_x\cos\theta + s_y\sin\theta$                          (A6)

then we can express the cylindrical dipole field as

$$
\boldsymbol{E}_{\text{int}} = \frac{-1}{2\varepsilon_0}\begin{bmatrix} s_x \\ s_y \end{bmatrix}
\tag{A7}
$$

$$
\boldsymbol{E}_{\text{ext}} = \frac{R^2}{2\varepsilon_0\rho^2}D\begin{bmatrix} s_x \\ s_y \end{bmatrix}
\tag{A8}
$$

where we introduce a matrix $D(\theta)$ defined as

$$
\begin{aligned}
D &= \begin{bmatrix} 2\cos^2\theta - 1 & 2\sin\theta\cos\theta \\ 2\sin\theta\cos\theta & 2\sin^2\theta - 1 \end{bmatrix} \\
&= \begin{bmatrix} \cos 2\theta & \sin 2\theta \\ \sin 2\theta & -\cos 2\theta \end{bmatrix}
\end{aligned}
\tag{A9}
$$

$D$ has the following property: Let $A$ be any matrix of the form

$$
A = \begin{bmatrix} a & -b \\ b & a \end{bmatrix}
\tag{A10}
$$

(Such a matrix combines a rotation with an isotropic scaling.) If $\hat{\boldsymbol{n}}$ (or $\hat{\boldsymbol{\rho}}$) is an outward unit vector $[\cos\theta, \sin\theta]^T$, then

$$
\hat{\boldsymbol{n}}^T DA = \hat{\boldsymbol{n}}^T A, \text{ whereas } \hat{\boldsymbol{n}}^T AD = \hat{\boldsymbol{n}}^T A^T.
\tag{A11}
$$

### Appendix B: Time dependence

We show in the main text that for a free-charge dipole $\sigma_{\text{free}}$ oriented towards $\hat{\boldsymbol{k}}$,

$$
\frac{\mathrm{d}}{\mathrm{d}t}\sigma_{\text{free}}\hat{\boldsymbol{k}} = -S\boldsymbol{E}_{\text{dip}}
\tag{B1}
$$

Now, $\sigma_{\text{net}}\hat{\boldsymbol{k}} = 2\varepsilon_0 \boldsymbol{E}_{\text{dip}}$ and $\sigma_{\text{free}} = \chi_{\text{e}}\sigma_{\text{net}}$, so the homogeneous behaviour of $\boldsymbol{E}_{\text{dip}}$ is

$$\frac{\mathrm{d}}{\mathrm{d}t}\boldsymbol{E}_{\text{dip}} = \frac{-S}{2\chi_{\text{e}}\varepsilon_0}\boldsymbol{E}_{\text{dip}} \tag{B2}$$

The forcing term in Eq. (25) determines the steady-state value of $\boldsymbol{E}_{\text{dip}}$, but the characteristic time $\tau$ required to *settle* on that value depends only on the inverse of the coefficient of this homogeneous term.

5  The matrix $S$ is of order $\sigma_{\text{P}}$ and the effective, low-frequency, perpendicular susceptibility of a magnetised plasma is $\chi_{\text{e}} = \rho_{\text{m}}/\varepsilon_0 B^2$, where $\rho_{\text{m}}$ is the plasma mass density. Hence

$$\tau \sim \frac{\chi_{\text{e}}\varepsilon_0}{\sigma_{\text{P}}} = \frac{(1+\kappa_{\text{i}}^2)}{\omega_{\text{i}}\kappa_{\text{i}}} = \frac{\csc\phi_{\text{i}}\sec\phi_{\text{i}}}{\omega_{\text{i}}} = \frac{\sec^2\phi_{\text{i}}}{\nu_{\text{in}}} \sim \frac{1}{\nu_{\text{in}}} \tag{B3}$$

So the time scale for the E-region alone to settle on a density-dependent drift speed is of the order of the mean time between ion-neutral collisions. The drift momentum of the F-region multiplies this by a further factor, equal to the ratio of F- to E-region integrated mass density, and this brings the time scale to the order of a second.

If the E-region is weak, such as in the polar cap without solar EUV, then the timescale will be longer, but still on the order of seconds. The underside of the F-region still has a reasonably high momentum transfer collision frequency.

The magnetospheric contribution to total flux-tube drift momentum also delays the structuring of the electric field as its momentum is used up by ionospheric ion-neutral collisions. The magnetospheric contribution of momentum is significantly more than the F-region's, but it arrives only after a transport delay due to Alfvén propagation of order 5 minutes. So the much larger magnetospheric drift momentum, making itself felt over so long a time, does not slow the approach to steady state as intensely as the F-region does. And even this momentum does not change the *steady-state* drift velocity ($\boldsymbol{E}_\perp$ structure) of the patch or depletion, which on closed field lines is determined by the (largely E-region) conductivity differences alone.

## Appendix C:  Momentum flux at boundaries

20 An objection might be raised that the jump in plasma speed at a boundary between regions of different density, e.g. across either of the two flat boundaries in the slab geometry, does not conserve momentum flux across the boundary. We have taken conservation of particle number into account in Eq. (4), but a careful reader may notice that in developing that equation we did not address the conservation of momentum flux across boundaries; however, implicit in our solutions is a flux of momentum from the magnetosphere into the ionosphere, and from there into the neutral gas.

25 We must orient ourselves and recall that in the ionosphere, plasma drift momentum is fleeting on the timescale of the momentum transfer collision frequencies $\nu_{\text{in}}$, and is passed on to the neutral gas. Ion drift, whether across these hypothetical sharp boundaries or in uniformity, is only maintained by the perpendicular electric field (in the frame of the neutrals) as it is in any conducting medium. Rather than being conserved *within the plasma* there is a continual flow of momentum from the plasma into the neutral gas, and the source of this momentum is the magnetospheric flow which generates the convection electric field in the ionosphere. But the FACs by which this background field is sustained can be occurring far outside the area of *our* study, at much higher and lower electric potentials – there is no requirement for them to be occurring on the boundaries

of the very patch we are studying, e.g. the Region 1 currents in the context of the polar cap, or the Region 1 and 2 currents in the context of the auroral oval.

Then one might further ask, if drift *momentum* is being drawn from the magnetosphere and deposited in the ionosphere by the FACs, which are carried by electrons of nearly negligible mass, how is this momentum transported *perpendicularly* to its direction of action, i.e. how is *moment* conserved? This is explained by the *torque* acting on a current loop (the ionospheric current closed by the FACs and depolarisation currents in the magnetosphere) within the geomagnetic field.

But now if a patch (or depletion) in the ionosphere is forced to convect at a different speed (as our model predicts) by these large-scale currents, there will be *new* stresses introduced in the plasma on the flux tube above the patch (or depletion) and FACs will be generated in the ionosphere, on the patch boundaries, that map this convection pattern upward to the magnetosphere. We wish to make it clear to the reader that we are not taking account of that time-dependent propagation. The balance in Eq. (4) and the electric field in Eqs. (26) and (27) assume that sufficient time has elapsed for Alfvén propagation to carry these features in the electric potential up into the magnetosphere as well, *if* the situation allow the patch to exist for a long enough time.

*Competing interests.*  None exist.

*Acknowledgements.*  The authors would like to acknowledge helpful contributions from K. Kabin and P. Perron.

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
