# Peer review of "On the convection of ionospheric density features"

_Annales Geophysicae, 2018_

## Referee Comment (RC1) · S. C. Buchert (Referee) · 31 Mar 2018

The authors present an analytic solution of a model for convecting ionospheric density structures. They make the point that the drift of the structures depends on the features of the structure, whether depletion or enhancement, and on how strong these are. And this could be relevant for a better understanding real world density structures in the Earth's ionosphere. The manuscript is very well written, I could follow the argumentations easily and enjoyed reading the paper. Proper credit is given to an earlier published similar result, though this was in quite a different context.

There would be quite a few objections, that the real ionosphere is more complicated and things are happening there differently than seen with the simplifying model. But

this would not be fair, the point is to develop an elegant model of convecting density structures and relate only a limited aspect of the result to the real ionosphere. However, even when taking this as a sort of Gedankenexperiment, we need to ask: is the model is physically consistent? I'm afraid, that I found a little bit of a spoiler there. The problem can be seen already with the slab model, Figure 1 and correspondig description in the text.

But first, in the E region recombination is effective, and a boundary with a density jump would have to be maintained by a corresponding jump in the production rate of ion-electron pairs. Equations 2 and 3 are only valid in the absence of production and recombination. Taking these into account can change the picture completely: the drift of a density structure would rather simply map the variation of the production rate in space and time (for example as produced by particle precipitation). However, I had promised not to object on grounds like "in the real world things are different". I and hopefully also the reader are willing to follow the authors: in the idealized E region of the model world recombination is switched off. Then a mass transport across the slab boundary needs to occur as described in the manuscript.

But then my alarm bells ring!: how about Newton, the conservation of momentum flux across the boundary? The jump of E across the boundary, required by Ohm's law, implies that the ExB component of the ion drift, the tangential $v_t$, also jumps, and any tangential acceleration experienced by crossing ions would need to be balanced by some force. The force could be magnetic stress. Then the from the magnetosphere well known jump condition is:

$$\rho_0 v_{n,0} v_{t,0} - B_{t,0} B_{n,0}/\mu_0 = \rho_1 v_{n,1} v_{t,1} - B_{t,1} B_{n,1}/\mu_0$$

(0 and 1 indicating the two sides of the boundary, n for normal direction, t for tangential).

Without magnetic stress, when $B_n = 0$, we have the jump condition of ordinary hydro-dynamics
$$\rho_0 v_{n,0} v_{t,0} = \rho_1 v_{n,1} v_{t,1}$$

In the manuscript the first jump condition, coming from the conservation of particle numbers, is used, but this second jump condition, coming from the conservation of momentum is ignored.

One can easily derive, taking into account equation 4 with $v_b = 0$, that $v_t$ cannot jump, $v_{t,0} = v_{t,1}$, if there is transport across the boundary, i.e. $v_n$ is different from zero. But $v_{t,0} = v_{t,1}$ would violate Ohm's law and current continuity. Or, allowing $v_t$ (and $E_n$) to jump, there cannot be mass transport across, $v_n$ must be zero.

Therefore, I'm afraid, that the model in its present form violates Newtons law, the conservation of momentum. Obviously this is the case not only for the slab, but also for the circular density structure with its complicated ion drift in and near the structure. By disallowing FACs, the model has no forces that could accomplish the derived pattern of particle motion.

Allowing for FACs and their closure would generate magnetic stress $B_n$. This can then produce a model that would be consistent with respect conservation of both mass and momentum, current continuity and Ohm's law. Whether such a model will then, at least qualitatively, still result in the convection of density structures as obtained in the manuscript, is not clear to me. It seems unlikely that an analytical treatment for the circular structure would be possible, but for the simple slab a fully consistent analytic solution (with non-zero $B_n$) should be achievable.

My objection of not conserving momentum does not disprove the conclusion of the authors about the convection of the density structure, but it questions whether the toilsome derivation in the manuscript really supports these conclusions. My objection is relatively fundamental, and I would insist that the problem needs to be admitted and discussed in a publication.

---

## Author Comment (AC1) · 11 Apr 2018

An objection might be raised that the jump in plasma speed at a boundary between regions of different density, e.g. across either of the two flat boundaries in the slab geometry, is non-physical in that it violate Newton's laws by not conserving momentum flux across the boundary. We appreciate the opportunity to explain something that may not have been dealt with well in our introduction.

We have taken conservation of particle number into account in Eq. (4), but a careful reader may notice that we have not addressed the conservation of momentum flux across any boundary, and indeed our solutions do not conserve momentum flux *within the ionosphere alone*.

[Figure]

First, we must orient ourselves and recall that in the ionosphere, plasma drift momentum is fleeting on the timescale of the momentum transfer collision frequencies $\nu_{\text{in}}$, and is passed on to the neutral gas. Ion drift, whether across these hypothetical sharp boundaries or in uniformity, is only maintained by the perpendicular electric field (in the frame of the neutrals) as it is in any conducting medium. Rather than being conserved *within the plasma* there is a continual flow of momentum from the plasma into the neutral gas, and the source of this momentum is the magnetospheric flow which generates the convection electric field in the ionosphere. But the FACs by which this field is sustained can be occurring far outside the area of *our* study, at much higher and lower electric potentials – there is no requirement for them to be occurring on the boundaries of the very patch we are studying.

We might then further ask, if drift *momentum* is being drawn from the magnetosphere and deposited in the ionosphere by the FACs, which are carried by electrons of nearly negligible mass, how is this momentum transported *perpendicularly* to its direction of action, i.e. how is *moment* conserved? This is explained by the *torque* acting on a current loop (the ionospheric current closed by the FACs and depolarisation currents in the magnetosphere) within the geomagnetic field.

Second, with this established background field and current in the ionosphere, a conductivity gradient would initially give rise to varied currents. But Ohm's law *per se* does not generate electric field structure. It produces charge accumulations and depletions. The electric field structure comes about from Maxwell's first law, $\epsilon_o \nabla \cdot \vec{E} = \rho_f - \nabla \cdot \vec{P}$. The RHS is the *net* charge density.

To see that such net charge densities occur, we can refer e.g. to SuperDARN maps, which show electric potential structure. The conditions being essentially magnetostatic, we have Poisson's equation, $\epsilon_o \nabla^2 \phi = -\rho_{net}$. This net charge density might suggest a problem with certain plasma instabilities, however it is *orders* of magnitude below the charge densities of the ions and electrons themselves.

Once this $\vec{E}$ field structure halts the accumulation, the currents reach steady state with $\nabla \cdot \vec{J} = 0$. In a 2-D plasma, there are no FACs. And we show in App. B that the magnetosphere is sufficiently far that it cannot provide FAC to efface $\vec{E}$ structure on sub-minute scales. Moreover, when an ionospheric structure has had time to begin drawing momentum from the magnetosphere, then it starts to deplete a finite supply, and begins to impress its structure on the magnetospheric flow.

We hope that this explanation serves to answer the concerns about the physicality of our results.

---

## Referee Comment (RC2) · Anonymous Referee #2 · 9 Sep 2018

General Comments

This paper addresses an important question in the high-latitude ionosphere, namely the convection of, and within, plasma density enhancements in the high-latitude ionosphere. There are some simplifications within the model that make the solution physically unrealistic, however I have no objections to these simplifications, provided that the work presented can help explain observations (see comments below). The work described in this paper is a necessary step towards a more realistic model. They are a sensible advance upon those used in previous publications and are likely to influence the development of more sophisticated models. The work is therefore likely to be of interest to researchers in the field and is, potentially, suitable for publication in Annales Geophysicae.

[Figure]

However, before I can recommend full publication, there are some changes that I believe would enhance this manuscript.

Specific Comments: Minor Changes

Section 1, line 22 (and elsewhere). The authors use the term "polar oval". Other authors have used 'polar cap', 'auroral oval' and 'polar region', but 'polar oval' is not one that I have encountered previously. Please could you define / change this term as appropriate?

Section 1.1, line 9 & section 4, line 24. The steepness of the boundaries modelled is unclear. In section 4, line 24 the authors say that they are studying "sharp, step boundaries", please add the word 'sharp' to section 1.1. Please also add typical values inside and outside the boundary, to allow for comparisons with other studies (see later comment). What is the basis for choosing these values? How do the densities modelled relate to the observations presented in the literature?

Section 3.1 (and elsewhere). Modelling an E-region patch. Polar cap patches are primarily an F-region phenomena. I am confused as to why the authors have modelled a patch in the E-region. Please would you either explain why the E-region has been used (with reference to observational studies in the published literature) or change the arguments to the F-region. Given the assumptions in the model, such a change should be a relatively minor alteration and I don't see that it would significantly affect the results obtained or the conclusions drawn.

Section 4.1. I found this section very confusing, and needed to re-read this several times. A diagram would significantly help with the clarity.

Specific Comments: Major Changes

The authors have assumed that the Earth's magnetic field, B, and the z-axis that they have defined are parallel (see, for example, Fig. 2), however this is only true at the geomagnetic pole. Given the other simplifications in this study, I'm not too worried

about the authors making this assumption, but you do need to discuss the effect of this assumption upon your results.

My largest objections to this paper at present is that there is not enough done to relate these results to other modelling & observational studies. Schunk & Sojka (1987) modelled a plasma density structure with a sharp boundary convecting over the polar cap and into the auroral oval. Please discuss how your work relates to, and builds on, this study. Moen et al. (2008) observed that the distribution of polar cap patches around magnetic midnight was asymmetric. Can the arguments which you have presented in section 3.5 explain this asymmetry? Oksavik et al. (2010) observed the rotation of a polar cap patch. Can the work which you have presented in section 3.4 explain this rotation? Numerous authors have presented theoretical or observational results that show small scale structures growing within polar cap patches, and have discussed why happens. I would suggest referring back to a selection of these results, to discuss how your paper relates to / advances our current understanding.

If your paper can explain observations, particularly those of Moen et al. (2008) and/or Oksavik et al. (2010), then this helps to justify the assumptions you have made. The model may not be physically realistic, but can be a step in the right direction towards a more realistic model. Such a discussion would show hoe your work would builds upon, and advances, our knowledge of this field.

I would like to state that I am not an author on any of the publications mentioned below.

Technical Comments

Section 4.1, line 13. The authors use the term 'FAC's'. Do you mean 'FACs'?

Writing style: Overall the writing style is good. The article is well-structured and clear. I do, however, have a dislike of sentences starting with the word 'And' (see section 1.2, line 26 and elsewhere). This is just be my personal view, but I think the paper would read better if these were rephrased (please ignore this comment If you wish).

References

Moen, J., N. Gulbrandsen, D. A. Lorentzen, and H. C. Carlson (2007), On the MLT distribution of F region polar cap patches at night, Geophys. Res. Lett., 34, L14113, doi:10.1029/2007GL029632.

Oksavik, K., V. L. Barth, J. Moen, and M. Lester (2010), On the entry and transit of high‐density plasma across the polar cap, J. Geophys. Res., 115, A12308, doi:10.1029/2010JA015817.

Schunk, R. W., and J. J. Sojka (1987), A theoretical study of the lifetime and transport of large ionospheric density structures, J. Geophys. Res., 92, 12,343–12,351, doi:10.1029/JA092iA11p12343.

---

## Author Comment (AC2) · 1 Oct 2018

**1   Reply**

The points raised are addressed below first, and following are several amended/added paragraphs for the manuscript. We have also prepared a revised manuscript with changes highlighted in red if it would be better to view the changes in that context

**1.1   Minor changes:**

We have corrected our terminology to 'polar region' and 'auroral oval.' We have added the word 'sharp' to the title and content of Sec. 1.1, however the idea applies to density steps of any value, and since we are not modeling but arguing generally we would like to avoid specific density values.

In Sec. 3.1, the E-region was named because the effect we are arguing comes about because of conductivity. Any structuring of the electric field will affect both the E- and F-regions equally, and if the E-region is uniformly weak, then it will be the F-region structure that determines the structuring of the electric field. The time required for the ionosphere to reach steady state will still be on the order of seconds, since the underside F-region, say below 200 km, still has reasonably high momentum transfer collision frequency. But then the ratio of integrated Pedersen to Hall conductivity will be very high, leading only to a simple case of the more general result we obtain for arbitrary $\kappa$. That explains our focus on the E-region. Our discussion has been expanded to explain why the results also apply to F-region patches.

A diagramme has been added to help explain the argument in Sec. 4.1 (now 4.2) about steepening gradient scale lengths.

The assumption of a vertical magnetic field has been mentioned, and also explained in the discussion.

**1.2   Major changes:**

The assumption of vertical magnetic field has been addressed in a new Sec. 4.1. The papers that the reviewer mentioned have been addressed, and indeed we think that they even offer evidence of the negative correlation we propose between plasma density and $\mathbf{E}$ field strength. A few other papers we mention also offer support. We address

modeling, citing the paper mentioned. We have not delved further into a review of modeling because a prescribed electric potential, or a fixed boundary condition for electric potential, is so common in those fields. We hope that the changes address the major concerns.

**1.3  Technical comments:**

By 'FAC's' we had intended 'FACs'; that has been changed. Several sentences beginning with 'And' have been reworked, and the remaining few we think are permissible.

**2  Amended/added paragraphs in the discussion:**

[revised manuscript text omitted]
_P - E_0 \sigma_H + k_y \sigma_H \\ -k_x \sigma_H + E_0 \sigma_P - k_y \sigma_P \end{bmatrix}' \cdot \hat{\boldsymbol{n}} \\
&= \cos\theta (-k_x \sigma_P - E_0 \sigma_H + k_y \sigma_H)' \\
&\quad + \sin\theta (-k_x \sigma_H + E_0 \sigma_P - k_y \sigma_P)'
\end{aligned} \tag{13}
$$

15    Using another result of App. A, at $\rho = R$,

$$\delta \boldsymbol{E}_{\text{ext}} = (k_x \cos\theta + k_y \sin\theta)\hat{\boldsymbol{\rho}} + (k_x \sin\theta - k_y \cos\theta)\hat{\boldsymbol{\theta}} \tag{14}$$

$$\boldsymbol{J}_{\text{ext}} = [\sigma] \begin{bmatrix} k_x \cos^2\theta + 2k_y \sin\theta \cos\theta - k_x \sin^2\theta \\ E_0 + 2k_x \sin\theta \cos\theta + k_y \sin^2\theta - k_y \cos^2\theta \end{bmatrix} \tag{15}$$

which after some straightforward steps yields a current *out of* the boundary of

$$
\begin{aligned}
J_{\text{ext}} &= E_0(\sigma_P \sin\theta - \sigma_H \cos\theta) + k_x \sigma_P \cos\theta \\

[revised manuscript text omitted]

We have taken conservation of particle number into account in Eq. (4), but a careful reader may notice that we have not

25    addressed the conservation of momentum flux across any boundary, and indeed our solutions do not conserve momentum flux *within the ionosphere alone*.

First, we must orient ourselves and recall that in the ionosphere, plasma drift momentum is fleeting on the timescale of the momentum transfer collision frequencies $\nu_{\text{in}}$, and is passed on to the neutral gas. Ion drift, whether across these hypothetical sharp boundaries or in uniformity, is only maintained by the perpendicular electric field (in the frame of the neutrals) as it

30    is in any conducting medium. Rather than being conserved *within the plasma* there is a continual flow of momentum from the plasma into the neutral gas, and the source of this momentum is the magnetospheric flow which generates the convection

electric field in the ionosphere. But the FACs by which this field is sustained can be occurring far outside the area of *our* study, at much higher and lower electric potentials – there is no requirement for them to be occurring on the boundaries of the very patch we are studying.

We might then further ask, if drift *momentum* is being drawn from the magnetosphere and deposited in the ionosphere by the FACs, which are carried by electrons of nearly negligible mass, how is this momentum transported *perpendicularly* to its direction of action, i.e. how is *moment* conserved? This is explained by the *torque* acting on a current loop (the ionospheric current closed by the FACs and depolarisation currents in the magnetosphere) within the geomagnetic field.

Second, with this established background field and current in the ionosphere, a conductivity gradient would initially give rise to varied currents. But Ohm's law *per se* does not generate electric field structure. It produces charge accumulations and depletions. The electric field structure comes about from Maxwell's first law, $\epsilon_o \nabla \cdot \boldsymbol{E} = \rho_f - \nabla \cdot \boldsymbol{P}$. The RHS is the *net* charge density.

To see that such net charge densities occur, we can refer e.g. to SuperDARN maps, which show electric potential structure. The conditions being essentially magnetostatic, we have Poisson's equation, $\epsilon_o \nabla^2 \phi = -\rho_{net}$. This net charge density might suggest a problem with certain plasma instabilities, however it is *orders* of magnitude below the charge densities of the ions and electrons themselves.

Once this $\boldsymbol{E}$ field structure halts the accumulation, the currents reach steady state with $\nabla \cdot \boldsymbol{J} = 0$. In a 2-D plasma there are no FACs, and we show in App. B that the magnetosphere is sufficiently far that it cannot provide FAC to efface $\boldsymbol{E}$ structure on sub-minute scales. Moreover, when an ionospheric structure has had time to begin drawing momentum from the magnetosphere, then it starts to deplete a finite supply, and begins to impress its structure on the magnetospheric flow.

**Appendix D: Response to referee #2 (not part of the manuscript)**

Changes discussed below have been made in the manuscript above and highlighted in red.

**D1   Minor changes:**

We have corrected our terminology to 'polar region' and 'auroral oval.' We have added the word 'sharp' to the title and content of Sec. 1.1, however the idea applies to density steps of any value, and since we are not modeling but arguing generally we would like to avoid specific density values.

In Sec. 3.1, the E-region was named because the effect we are arguing comes about because of conductivity. Any structuring of the electric field will affect both the E- and F-regions equally, and if the E-region is uniformly weak, then it will be the F-region structure that determines the structuring of the electric field. The time required for the ionosphere to reach steady state will still be on the order of seconds, since the underside F-region, say below 200 km, still has reasonably high momentum transfer collision frequency. But then the ratio of integrated Pedersen to Hall conductivity will be very high, leading only to a simple case of the more general result we obtain for arbitrary $\kappa$. That explains our focus on the E-region. Our discussion has been expanded to explain why the results also apply to F-region patches.

A diagramme has been added to help explain the argument in Sec. 4.2 (formerly 4.1) about steepening gradient scale lengths. The assumption of a vertical magnetic field has been mentioned, and also explained in the discussion.

**D2   Major changes:**

The assumption of vertical magnetic field has been addressed in Sec. 4.1. The papers that the reviewer mentioned have been
5   addressed, and indeed we think that they even offer evidence of the negative correlation we propose between plasma density and $E$ field strength. A few other papers we mention also offer support. We address modeling, citing the paper mentioned. We have not delved further into a review of modeling because a prescribed electric potential, or a fixed boundary condition for electric potential, is so common in those fields. We hope that the changes address the major concerns.

**D3   Technical comments:**

10   By 'FAC's' we had intended 'FACs'; that has been changed. Several sentences beginning with 'And' have been reworked, and the remaining few we think are permissible.

*Competing interests.*   None exist.

*Acknowledgements.*   The authors would like to acknowledge helpful contributions from K. Kabin and P. Perron.

---

## Author Response (AR1)

[revised manuscript text omitted]
_{\mathrm{int}} &= \begin{bmatrix} -k_x\sigma_{\mathrm{P}} - E_0\sigma_{\mathrm{H}} + k_y\sigma_{\mathrm{H}} \\ -k_x\sigma_{\mathrm{H}} + E_0\sigma_{\mathrm{P}} - k_y\sigma_{\mathrm{P}} \end{bmatrix}' \cdot \hat{\boldsymbol{n}} \\ &= \cos\theta(-k_x\sigma_{\mathrm{P}} - E_0\sigma_{\mathrm{H}} + k_y\sigma_{\mathrm{H}})' \\ &\quad + \sin\theta(-k_x\sigma_{\mathrm{H}} + E_0\sigma_{\mathrm{P}} - k_y\sigma_{\mathrm{P}})' \end{aligned} \tag{13}$$

Using another result of App. A, at $\rho = R$,

20    $$\delta\boldsymbol{E}_{\mathrm{ext}} = (k_x\cos\theta + k_y\sin\theta)\hat{\boldsymbol{\rho}} + (k_x\sin\theta - k_y\cos\theta)\hat{\boldsymbol{\theta}} \tag{14}$$

$$\boldsymbol{J}_{\mathrm{ext}} = [\sigma] \begin{bmatrix} k_x\cos^2\theta + 2k_y\sin\theta\cos\theta - k_x\sin^2\theta \\ E_0 + 2k_x\sin\theta\cos\theta + k_y\sin^2\theta - k_y\cos^2\theta \end{bmatrix} \tag{15}$$

which after some straightforward steps yields a current *out of* the boundary of

$$J_{\mathrm{ext}} = E_0(\sigma_{\mathrm{P}}\sin\theta - \sigma_{\mathrm{H}}\cos\theta) + k_x\sigma_{\mathrm{P}}\cos\theta$$

25    $$-k_x\sigma_{\mathrm{H}}\sin\theta + k_y\sigma_{\mathrm{P}}\sin\theta + k_y\sigma_{\mathrm{H}}\cos\theta \tag{16}$$

Setting the cosine terms in $J_{\text{int}}$ and $J_{\text{ext}}$ equal,

$$(\sigma_P + \sigma_P')k_x + (\sigma_H - \sigma_H')k_y = E_0(\sigma_H - \sigma_H') \tag{17}$$

and setting the sine terms equal,

$$(-\sigma_H + \sigma_H')k_x + (\sigma_P + \sigma_P')k_y = E_0(-\sigma_P + \sigma_P') \tag{18}$$

5    These two linear equations in $k_x$ and $k_y$ are

$$\begin{bmatrix} \sigma_P + \sigma_P' & \sigma_H - \sigma_H' \\ -\sigma_H + \sigma_H' & \sigma_P + \sigma_P' \end{bmatrix} \begin{bmatrix} k_x \\ k_y \end{bmatrix} = E_0 \begin{bmatrix} \sigma_H - \sigma_H' \\ -\sigma_P + \sigma_P' \end{bmatrix} \tag{19}$$

$$\Big[ [\sigma]' + [\sigma]^T \Big] \begin{bmatrix} k_x \\ k_y \end{bmatrix} = \Big[ [\sigma]' - [\sigma] \Big] \begin{bmatrix} 0 \\ E_0 \end{bmatrix} \tag{20}$$

The LHS represents the rate of loss of net charge from the dipole $\sigma = 2\varepsilon_0 E_{\text{dip}}$. On the RHS, the *difference* in $[\sigma]$ between the
10    patch and its surroundings is forcing its polarisation. Hence we can write

$$S\boldsymbol{E}_{\text{dip}} = (n-1)[\sigma]\boldsymbol{E}_0 \tag{21}$$

where the matrix on the LHS is

$$S = \begin{bmatrix} \sigma_P(n+1) & \sigma_H(-n+1) \\ \sigma_H(n-1) & \sigma_P(n+1) \end{bmatrix} \tag{22}$$

which is *nearly* scalar for $n \approx 1$, but let us define $H$ by writing

15    $$S = (n+1)\sigma_P \begin{bmatrix} 1 & -\eta\sigma_H/\sigma_P \\ \eta\sigma_H/\sigma_P & 1 \end{bmatrix} = (n+1)\sigma_P H \tag{23}$$

$$(n+1)\sigma_P H \boldsymbol{E}_{\text{dip}} = (n-1)[\sigma]\boldsymbol{E}_0 \tag{24}$$

Solving for $\boldsymbol{E}_{\text{dip}}$, a polarisation with

$$\boldsymbol{E}_{\text{dip}} = \frac{\eta}{\sigma_P} H^{-1}[\sigma]\boldsymbol{E}_0 \tag{25}$$

20    yields a divergence-free current field.

The steady-state electric field in and around the patch, using Eqs. (A7) and (A8), is

$$\boldsymbol{E}_{\text{int}} = \left( I - \frac{\eta}{\sigma_P} H^{-1}[\sigma] \right) \boldsymbol{E}_0 \tag{26}$$

[Figure]

**Figure 4.** The Hall current, not yet shown in Fig. 2, adds a rotation to the polarisation of the patch, however the exact solution for a sharp circular boundary is still a cylindrical dipole.

$$\boldsymbol{E}_{\text{ext}} = \left( I + \frac{\eta R^2}{\sigma_{\text{P}} \rho^2} D(\theta) H^{-1}[\sigma] \right) \boldsymbol{E}_0 \tag{27}$$

From this expression for $\boldsymbol{E}_{\text{int}}$, one can verify that it is at an angle relative to $\boldsymbol{E}_0$ whose tangent is $\eta \sigma_{\text{H}} / \sigma_{\text{P}}$. This agrees with Hysell and Drexler's Eq. (9), which gives us confidence in our results, although we focus below on the *boundary*'s drift, rather than that of the ions inside.

**3.2 Simplifying assumptions**

We shall deal here with a single ion species, and assume that electrons are fully magnetised. These assumptions are not necessary for a unique solution, but will greatly simplify the algebra. Under these assumptions, and using the ion magnetisation parameter $\kappa_{\text{i}} = \omega_{\text{i}} / \nu_{\text{in}}$, we have

$$\sigma_{\text{P}} = \frac{q_i n_i}{B} \left( \frac{\kappa_{\text{i}}}{1 + \kappa_{\text{i}}^2} \right) \tag{28}$$

$$\sigma_{\text{H}} = \frac{q_i n_i}{B} \left( \frac{1}{1 + \kappa_{\text{i}}^2} \right) \tag{29}$$

Thus $\sigma_{\text{P}} = \kappa_{\text{i}} \sigma_{\text{H}}$, and $|\kappa_{\text{e}}|$ is much larger than both $\kappa_{\text{
[revised manuscript text omitted]

10 We have taken conservation of particle number into account in Eq. (4), but a careful reader may notice that we have not addressed the conservation of momentum flux across any boundary, and indeed our solutions do not conserve momentum flux *within the ionosphere alone*.

First, we must orient ourselves and recall that in the ionosphere, plasma drift momentum is fleeting on the timescale of the momentum transfer collision frequencies $\nu_{in}$, and is passed on to the neutral gas. Ion drift, whether across these hypothetical

15 sharp boundaries or in uniformity, is only maintained by the perpendicular electric field (in the frame of the neutrals) as it is in any conducting medium. Rather than being conserved *within the plasma* there is a continual flow of momentum from the plasma into the neutral gas, and the source of this momentum is the magnetospheric flow which generates the convection electric field in the ionosphere. But the FACs by which this field is sustained can be occurring far outside the area of *our* study, at much higher and lower electric potentials – there is no requirement for them to be occurring on the boundaries of the very

20 patch we are studying.

We might then further ask, if drift *momentum* is being drawn from the magnetosphere and deposited in the ionosphere by the FACs, which are carried by electrons of nearly negligible mass, how is this momentum transported *perpendicularly* to its direction of action, i.e. how is *moment* conserved? This is explained by the *torque* acting on a current loop (the ionospheric current closed by the FACs and depolarisation currents in the magnetosphere) within the geomagnetic field.

25 Second, with this established background field and current in the ionosphere, a conductivity gradient would initially give rise to varied currents. But Ohm's law *per se* does not generate electric field structure. It produces charge accumulations and depletions. The electric field structure comes about from Maxwell's first law, $\epsilon_o \nabla \cdot E = \rho_f - \nabla \cdot P$. The RHS is the *net* charge density.

To see that such net charge densities occur, we can refer e.g. to SuperDARN maps, which show electric potential structure.

30 The conditions being essentially magnetostatic, we have Poisson's equation, $\epsilon_o \nabla^2 \phi = -\rho_{net}$. This net charge density might suggest a problem with certain plasma instabilities, however it is *orders* of magnitude below the charge densities of the ions and electrons themselves.

Once this $E$ field structure halts the accumulation, the currents reach steady state with $\nabla \cdot J = 0$. In a 2-D plasma there are no FACs, and we show in App. B that the magnetosphere is sufficiently far that it cannot provide FAC to efface $E$ structure on sub-minute scales. Moreover, when an ionospheric structure has had time to begin drawing momentum from the magnetosphere, then it starts to deplete a finite supply, and begins to impress its structure on the magnetospheric flow.

**5 Appendix D: Response to referee #2 (not part of the manuscript)**

Changes discussed below have been made in the manuscript above and highlighted in red.

**D1 Minor changes:**

We have corrected our terminology to 'polar region' and 'auroral oval.' We have added the word 'sharp' to the title and content of Sec. 1.1, however the idea applies to density steps of any value, and since we are not modeling but arguing generally we would like to avoid specific density values.

In Sec. 3.1, the E-region was named because the effect we are arguing comes about because of conductivity. Any structuring of the electric field will affect both the E- and F-regions equally, and if the E-region is uniformly weak, then it will be the F-region structure that determines the structuring of the electric field. The time required for the ionosphere to reach steady state will still be on the order of seconds, since the underside F-region, say below 200 km, still has reasonably high momentum transfer collision frequency. But then the ratio of integrated Pedersen to Hall conductivity will be very high, leading only to a simple case of the more general result we obtain for arbitrary $\kappa$. That explains our focus on the E-region. Our discussion has been expanded to explain why the results also apply to F-region patches.

A diagramme has been added to help explain the argument in Sec. 4.2 (formerly 4.1) about steepening gradient scale lengths.

The assumption of a vertical magnetic field has been mentioned, and also explained in the discussion.

**D2 Major changes:**

The assumption of vertical magnetic field has been addressed in Sec. 4.1. The papers that the reviewer mentioned have been addressed, and indeed we think that they even offer evidence of the negative correlation we propose between plasma density and $E$ field strength. A few other papers we mention also offer support. We address modeling, citing the paper mentioned. We have not delved further into a review of modeling because a prescribed electric potential, or a fixed boundary condition for electric potential, is so common in those fields. We hope that the changes address the major concerns.

**D3 Technical comments:**

By 'FAC's' we had intended 'FACs'; that has been changed. Several sentences beginning with 'And' have been reworked, and the remaining few we think are permissible.

*Competing interests.* None exist.

*Acknowledgements.* The authors would like to acknowledge helpful contributions from K. Kabin and P. Perron.

---

## Editor Decision (ED1)

Dear Authors,

Thank you for sending us your improved article and giving us the chance to consider your work. Your article was read by me and two external reviewers. We enjoyed reading your article and are pleased to make a provisional offer of publication if you are able to revise the paper to address the following additional comments of one of the reviewers:

*Reviewer's comments to the revised version*

The authors have added text in sections "4.1 Addressing some idealisations" and "4.3 Search for observational support" and made further minor changes. Disappointingly for me, none of the additions and modification seem to address the points that I had made in my first comment. In their reply the authors seem to admit that there are limitations to their approach, particularly that momentum is not conserved in their model, but then they provide a lengthy text which I have a hard time to understand in parts. The manuscript would greatly benefit, if limitations and assumptions were properly stated. These are:

1) Equations (3) and consequently (4) is only correct in the absence of any ionization/recombination. In the Earth's E region, where the model is supposed to be applicable, ionization and recombination are very significant terms on the continuity equation. The assumption of no ionization/recombination is still nowhere stated in the manuscript.

2) Momentum is not conserved by the model, because FACs are suppressed. The statement "...there is no requirement for FACs to be occurring on the boundaries of the very patch..." is not correct. The momentum balance could be established by magnetic stress as the first equation of my comment shows for the moving 1d interface. To get the magnetic field jump across sharp interfaces, local FACs are needed. FACs "occurring far outside the area of our study" do not help for the local momentum balance.

I had not suggested that the model is reworked to fix these limitations, only that they are properly exposed also to the reader.

Kindest regards

Yours cordially
Dalia Buresova

---

## Author Response (AR2)

Dr. Buresova,

Below is a revision which we hope will satisfy the remaining concerns of the reviewer. The changes are in red. It is also uploaded in another pdf with all black text. The changes occur in sections 1, 1.1, the title of 1.2, the end of the conclusions in 5, and App. C. While in some cases the changes attempt to further explain or justify the assumptions we have made, they are mostly aimed at clarifying what assumptions we have made and highlighting them for the reader.

Our explanation in App. C about momentum conservation was admittedly not clear enough in our first revision. It is conserved, but we wanted to explain that the momentum that is passed on to the neutral gas is being drawn from the magnetospheric convection. In a full 3-D model there would be another momentum flow on the field lines above the patch, but I think we have stated there (as also elsewhere in the text) the limitations of our model.

The assumption of no ionisation/recombination has now been stated in section 1.1.

The reviewer's comment 2) refers to "the first equation of my comments." I regret that we were not able to identify an equation in the comments, or whether they intended our equation (3), which was the first one addressed in their comments. I think that by mentioning in several places including the abstract that our model assumes a 2-D plasma, that we address our lack of treatment of the FACs on the boundary of the patch. A more detailed treatment of cause and effect could look at the changes in magnetic stress but the net result, as he states as well, is the FAC generation on the patch boundary that we have stated (I hope clearly enough) is beyond the scope of our paper.

We respect the reviewer's emphasis on highlighting the assumptions more clearly, and I hope that the changes we have made will convey those limitations more openly.

My delay in replying was due to communication with my co-authors and our various duties, which I appreciate we all have. Thanks again to you and the reviewers for your long-suffering efforts in bringing our paper closer to publication.

J. de Boer

[revised manuscript text omitted]
_{\text{P}} \begin{bmatrix} 1 & -\eta\sigma_{\text{H}}/\sigma_{\text{P}} \\ \eta\sigma_{\text{H}}/\sigma_{\text{P}} & 1 \end{bmatrix} = (n+1)\sigma_{\text{P}}H \tag{23}$$

$$(n+1)\sigma_{\text{P}}H\boldsymbol{E}_{\text{dip}} = (n-1)[\sigma]\boldsymbol{E}_0 \tag{24}$$

Solving for $\boldsymbol{E}_{\text{dip}}$, a polarisation with

$$\boldsymbol{E}_{\text{dip}} = \frac{\eta}{\sigma_{\text{P}}}H^{-1}[\sigma]\boldsymbol{E}_0 \tag{25}$$

20   yields a divergence-free current field.

The steady-state electric field in and around the patch, using Eqs. (A7) and (A8), is

$$\boldsymbol{E}_{\text{int}} = \left( I - \frac{\eta}{\sigma_{\text{P}}}H^{-1}[\sigma] \right) \boldsymbol{E}_0 \tag{26}$$

[Figure]

**Figure 4.** The Hall current, not yet shown in Fig. 2, adds a rotation to the polarisation of the patch, however the exact solution for a sharp circular boundary is still a cylindrical dipole.

$$\boldsymbol{E}_{\text{ext}} = \left( I + \frac{\eta R^2}{\sigma_{\text{P}} \rho^2} D(\theta) H^{-1}[\sigma] \right) \boldsymbol{E}_0 \tag{27}$$

From this expression for $\boldsymbol{E}_{\text{int}}$, one can verify that it is at an angle relative to $\boldsymbol{E}_0$ whose tangent is $\eta \sigma_{\text{H}}/\sigma_{\text{P}}$. This agrees with Hysell and Drexler's Eq. (9), which gives us confidence in our results, although we focus below on the *boundary*'s drift, rather than that of the ions inside.

**3.2 Simplifying assumptions**

We shall deal here with a single ion species, and assume that electrons are fully magnetised. These assumptions are not necessary for a unique solution, but will greatly simplify the algebra. Under these assumptions, and using the ion magnetisation parameter $\kappa_{\text{i}} = \omega_{\text{i}}/\nu_{\text{in}}$, we have

$$\sigma_{\text{P}} = \frac{q_i n_i}{B} \left( \frac{\kappa_{\text{i}}}{1 + \kappa_{\text{i}}^2} \right) \tag{28}$$

$$\sigma_{\text{H}} = \frac{q_i n_i}{B} \left( \frac{1}{1 + \kappa_{\text{i}}^2} \right) \tag{29}$$

Thus $\sigma_{\text{P}} = \kappa_{\text{i}} \sigma_{\text{H}}$, and $|\kappa_{\text{e}}|$ is much larger than both $\kappa_{\text{
[revised manuscript text omitted]